# Distinct mesoderm migration phenotypes in extra-embryonic and embryonic regions of the early mouse embryo

**Bechara Saykali[1], Navrita Mathiah[1†], Wallis Nahaboo[1†], Marie-Lucie Racu[1], Latifa Hammou[1], Matthieu Defrance[2], Isabelle Migeotte[1,3]***

[1]IRIBHM, Université Libre de Bruxelles, Brussels, Belgium; [2]Interuniversity Institute of Bioinformatics in Brussels, Université Libre de Bruxelles, Brussels, Belgium; [3]Walloon Excellence in Lifesciences and Biotechnology, Wallonia, Belgium

**Abstract** In mouse embryo gastrulation, epiblast cells delaminate at the primitive streak to form mesoderm and definitive endoderm, through an epithelial-mesenchymal transition. Mosaic expression of a membrane reporter in nascent mesoderm enabled recording cell shape and trajectory through live imaging. Upon leaving the streak, cells changed shape and extended protrusions of distinct size and abundance depending on the neighboring germ layer, as well as the region of the embryo. Embryonic trajectories were meandrous but directional, while extra-embryonic mesoderm cells showed little net displacement. Embryonic and extra-embryonic mesoderm transcriptomes highlighted distinct guidance, cytoskeleton, adhesion, and extracellular matrix signatures. Specifically, intermediate filaments were highly expressed in extra-embryonic mesoderm, while live imaging for F-actin showed abundance of actin filaments in embryonic mesoderm only. Accordingly, *Rhoa* or *Rac1* conditional deletion in mesoderm inhibited embryonic, but not extra-embryonic mesoderm migration. Overall, this indicates separate cytoskeleton regulation coordinating the morphology and migration of mesoderm subpopulations.
DOI: https://doi.org/10.7554/eLife.42434.001

**\*For correspondence:**
imigeott@ulb.ac.be

[†]These authors contributed equally to this work

**Competing interests:** The authors declare that no competing interests exist.

## Introduction

In mice, a first separation of embryonic and extra-embryonic lineages begins in the blastocyst at embryonic day (E) 3.5 when the trophectoderm is set aside from the inner cell mass. A second step is the segregation of the inner cell mass into the epiblast, the precursor of most fetal cell lineages, and the extra-embryonic primitive endoderm (*Chazaud and Yamanaka, 2016*). At E6, the embryo is cup-shaped and its anterior-posterior axis is defined. It comprises three cell types, arranged in two layers: the inner layer is formed by epiblast, distally, and extra-embryonic ectoderm, proximally; the outer layer, visceral endoderm, covers the entire embryo surface. The primitive streak, site of gastrulation, is formed at E6.25 in the posterior epiblast, at the junction between embryonic and extra-embryonic regions, and subsequently elongates to the distal tip of the embryo. The primitive streak is the region of the embryo where epiblast cells delaminate through epithelial-mesenchymal transition to generate a new population of mesenchymal cells that form the mesoderm and definitive endoderm layers.

All mesoderm, including the extra-embryonic mesoderm, is of embryonic epiblast origin. At the onset of gastrulation, emerging mesoderm migrates either anteriorly as so-called embryonic mesodermal wings, or proximally as extra-embryonic mesoderm (*Arnold and Robertson, 2009*; *Sutherland, 2016*). Cell lineages studies showed that there is little correlation between the position of mesoderm progenitors in the epiblast and the final localization of mesoderm descendants (*Lawson et al., 1991*). Rather, the distribution of mesoderm subpopulations depends on the

**eLife digest** As an embryo develops, its cells divide and specialize to form different tissues and organs. Early in development the cells arrange into so-called germ layers, which each produce particular types of tissue. One of these layers, called the mesoderm, develops into the muscles, bones and circulatory system of the embryo. It also contributes to the support structures that feed and protect the embryo, such as the placenta, umbilical cord and yolk sac. If these 'extra-embryonic' structures do not develop correctly, the embryo may not grow properly.

Much of what we know about how the cells of the mesoderm move around to form different tissues comes from studies of species that lay eggs; for example, chicks, frogs and fish. The initial steps of embryo development in these animals are similar to how mammals develop, but bigger differences emerge as the extra-embryonic tissues start to form. Recent methodological advances are now making it possible to dynamically study this later stage of development in live mouse embryos.

Saykali et al. studied mouse embryos whose mesoderm cells contained a 'reporter' that allowed them to be identified when viewed using a microscopy technique known as two-photon live imaging. This approach allows cells to be tracked as they move through living tissue. Saykali et al. found that the mesoderm cells change shape depending on which region of the embryo they are in, and on which germ layer they are next to. The cells that become extra-embryonic are larger and longer, and develop small protrusions. Instead of moving directly to their destinations, they tend to zigzag.

Further experiments revealed that embryonic and extra-embryonic mesoderm cells produce different amounts of several proteins, including the distinct types of filaments that act as the cell's internal skeleton. Mesoderm cells that are destined to become extra-embryonic depend less on signaling proteins called Rho GTPases to move around.

Knowing how mesoderm cells form extra-embryonic structures will help researchers to understand how problems with these structures can affect how embryos grow. The techniques used by Saykali et al. will also help to design new ways to cultivate mesoderm cells in the laboratory for future experiments. These could, for example, investigate whether human mesoderm cells develop in the same way as mice mesoderm cells.

DOI: https://doi.org/10.7554/eLife.42434.002

temporal order and anterior-posterior location of cell recruitment through the primitive streak (*Kinder et al., 1999*). Posterior primitive streak cells are the major source of extra-embryonic mesoderm, while cells from middle and anterior primitive streak are mostly destined to the embryo proper. However, there is overlap of fates between cells delaminating at different sites and timings (*Kinder et al., 1999*; *Kinder et al., 2001*). Extra-embryonic mesoderm contributes to the amnion, allantois, chorion, and visceral yolk sac. It has important functions in maternal-fetal protection and communication, as well as in primitive erythropoiesis (*Watson and Cross, 2005*). Embryonic mesoderm separates into lateral plate, intermediate, paraxial and axial mesoderm, and ultimately gives rise to cranial and cardiac mesenchyme, blood vessels and hematopoietic progenitors, urogenital system, muscles and bones, among others. Endoderm precursors co-migrate with mesoderm progenitors in the wings and undergo a mesenchymal-epithelial transition to intercalate into the visceral endoderm (*Viotti et al., 2014*).

Mesoderm migration mechanisms have mostly been studied in fly, fish, frog and chicken embryos. During fly gastrulation, mesodermal cells migrate as a collective (*Bae et al., 2012*). In the fish *Fundulus heteroclitus*, deep cells of the dorsal germ ring move as loose clusters with meandering trajectories (*Trinkaus et al., 1992*). At mid-gastrulation, zebrafish lateral mesoderm cells are not elongated and migrate as individuals along indirect paths, while by late gastrulation, cells are more polarized and their trajectories are straighter, resulting in higher speed (*Jessen et al., 2002*). In zebrafish prechordal plate, all cells have similar migration properties but they require contact between each other for directional migration (*Dumortier et al., 2012*). In chick, cells migrate in a very directional manner at high density. Cells are continually in close proximity, even though they frequently make and break contacts with their neighbors (*Chuai et al., 2012*).

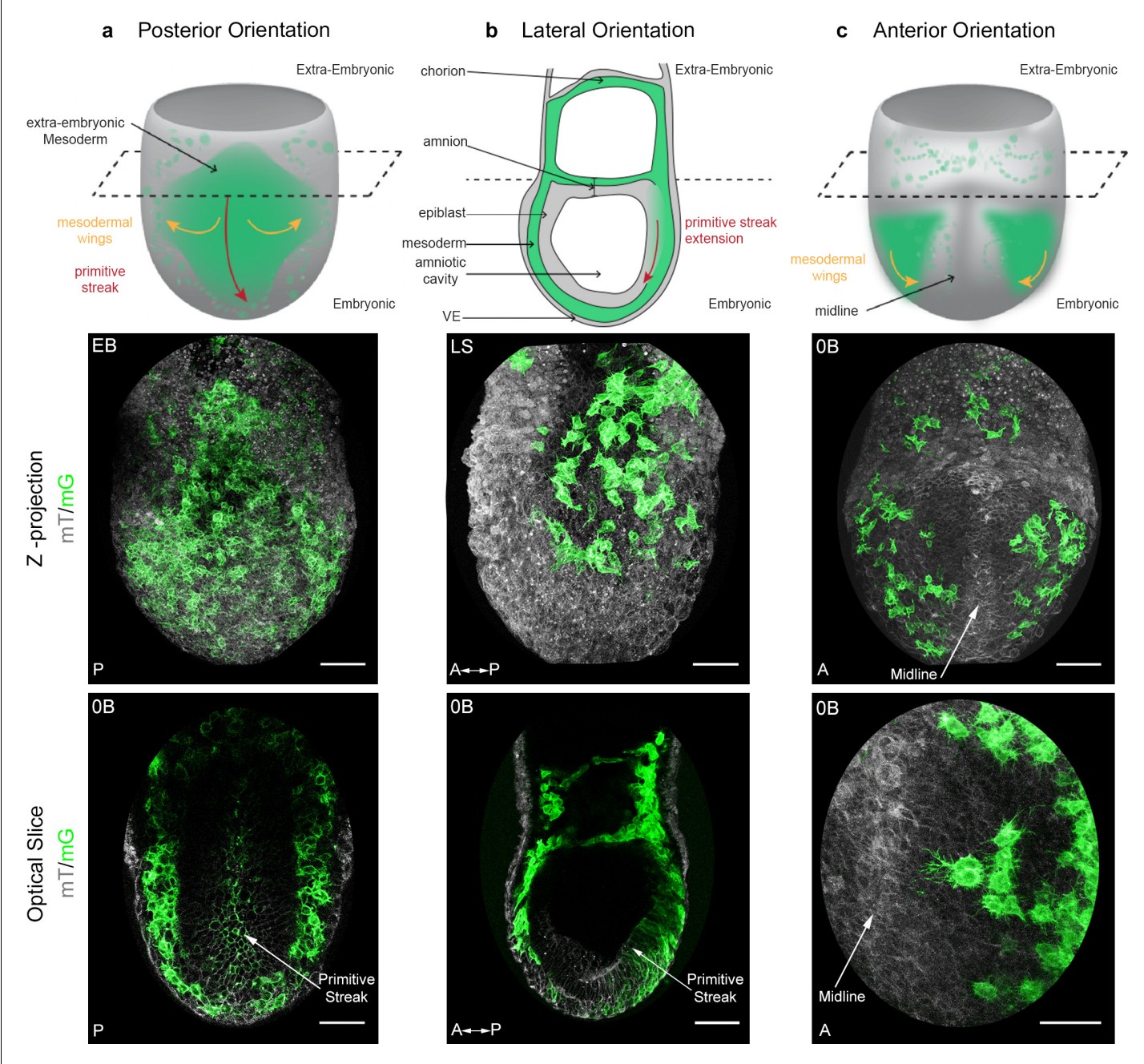

**Figure 1.** Mosaic membrane GFP labeling of nascent mesoderm allows following individual cell migration through embryo live imaging. (**a**) Posterior view. Top: 3D scheme with mesoderm layer in green and the rest of the embryo in grey. The dashed line separates embryonic and extra-embryonic regions. Middle: Z-projection of two-photon stack. Bottom: optical slice highlighting the primitive streak. (**b**) Lateral view, anterior to the left. Top: 2D scheme. Middle: Z-projection of two-photon stack showing cells progression from posterior to anterior. Bottom: sagittal optical slice. (**c**) Anterior view. Top: 3D scheme. Middle: Z-projection of two-photon stack with most anterior cells reaching the midline. Bottom: optical slice zoomed on filopodia extending towards the midline. All embryos were dissected at E7.25 and are at Late Streak/Early Bud stage. VE: Visceral Endoderm; mG: membrane GFP, in green; mT: membrane dtTomato, in grey; EB: Early Bud; LS: Late Streak; 0B: Zero Bud. (Scale bars: 50 µm).
DOI: https://doi.org/10.7554/eLife.42434.003

The following source data and figure supplements are available for figure 1:

**Figure supplement 1.** Live imaging of *Brachyury (T)*-Cre; mTmG embryos.
DOI: https://doi.org/10.7554/eLife.42434.004

**Figure supplement 1—source data 1.** Division events in mesoderm cells.
DOI: https://doi.org/10.7554/eLife.42434.005

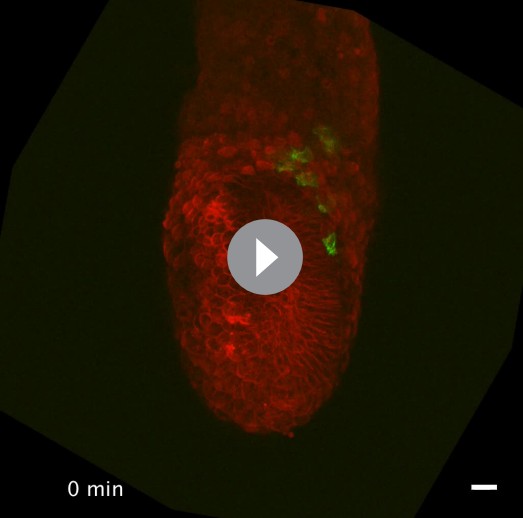

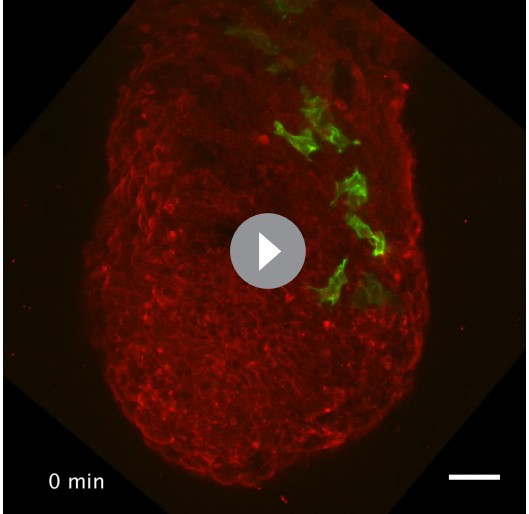

**Video 1.** Mesoderm cells migrating towards extra-embryonic and embryonic regions. Z-projections of confocal stacks from a *T*-Cre; mTmG embryo dissected at E6.75 (Early Streak stage) and imaged for 320 min. Mesoderm cells express membrane GFP (green); all other cells express membrane dtTomato (red). Anterior oblique orientation with posterior to the right (scale bar: 50 μm).
DOI: https://doi.org/10.7554/eLife.42434.006

**Video 2.** 'Trial and error' trajectories. Z-projections of confocal stacks from a *T*-Cre; mTmG embryo dissected at E6.75 (Mid Streak stage) and imaged for 260 min. Mesoderm cells express membrane GFP (green); all other cells express membrane dtTomato (red). Anterior oblique orientation with posterior to the right (scale bar: 50 μm).
DOI: https://doi.org/10.7554/eLife.42434.007

Relatively little is known about mesoderm migration in mice because most mutant phenotypes with mesodermal defects result from anomalies in primitive streak formation, mesoderm specification, or epithelial-mesenchymal transition (*Arnold and Robertson, 2009*), precluding further insight into cell migration mechanisms. We previously identified a role for the Rho GTPase Rac1, a mediator of cytoskeletal reorganization, in mesoderm migration and adhesion (*Migeotte et al., 2011*).

Recent advances in mouse embryo culture and live imaging have overcome the challenge of maintaining adequate embryo growth and morphology while performing high-resolution imaging. It facilitated the uncovering of the precise spatial and temporal regulation of cellular processes and disclosed that inaccurate conclusions had sometimes been drawn from static analyses (*Viotti et al., 2014*). Live imaging of mouse embryos bearing a reporter for nuclei has pointed towards individual rather than collective migration in the mesodermal wings (*Ichikawa et al., 2013*). Very recently, a spectacular adaptive light sheet imaging approach allowed reconstructing fate maps at the single cell level from gastrulation to early organogenesis (*McDole et al., 2018*). However, little is known about how mesoderm populations regulate their shape and migration mechanisms as they travel across distinct embryo regions to fulfill their respective fates.

Here, high-resolution live imaging of nascent mesoderm expressing membrane-bound GFP was used to define the dynamics of mesoderm cell morphology and its trajectories. Mesoderm cells exhibited a variety of cell shape changes determined by their spatial localization in the embryo, and the germ layer they were in contact with. The embryonic mesoderm migration path was meandrous but directional, and depended on the Rho GTPases Rhoa and Rac1. Extra-embryonic mesoderm movement was, strikingly, GTPases independent. Transcriptomes of different mesoderm populations uncovered specific sets of guidance, adhesion, cytoskeleton and matrix components, which may underlie the remarkable differences in cell behavior between mesoderm subtypes.

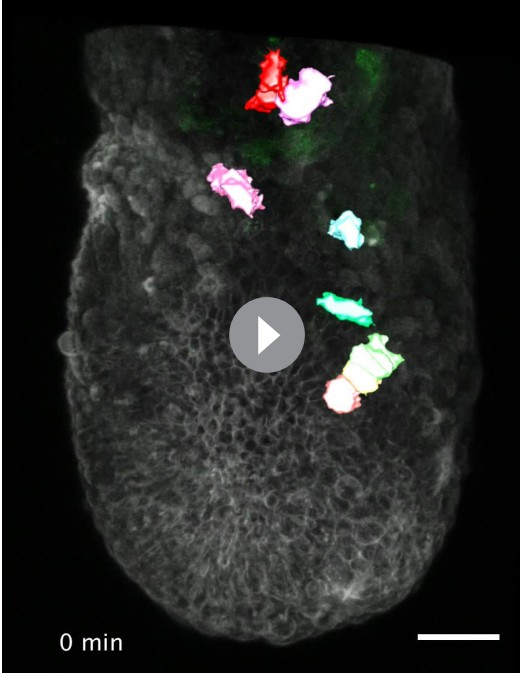

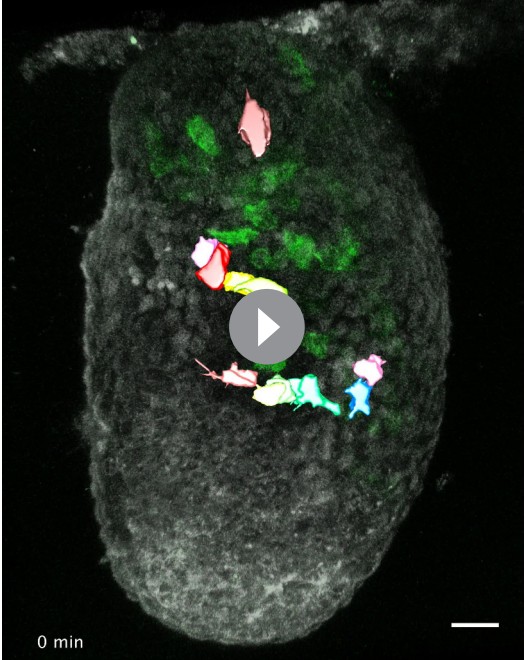

**Video 3.** Tracking mesoderm migration. 3D snapshots of confocal stacks from a *T*-Cre; mTmG embryo dissected at E6.75 (Early Streak stage) and imaged for 180 min, with manually highlighted cells tracked throughout the time lapse. Video shows highlighted cells first, then the original images (membrane GFP, in green) in a looping fashion for comparison. All other cells express membrane dtTomato (grey). Lateral orientation with anterior to the left (scale bar: 50 μm).
DOI: https://doi.org/10.7554/eLife.42434.008

**Video 4.** Tracking mesoderm migration. 3D snapshots of confocal stacks from a *T*-Cre; mTmG embryo dissected at E6.75 (Early Streak stage) and imaged for 160 min, with manually highlighted cells tracked throughout the time lapse. Video shows highlighted cells first, then the original images (membrane GFP, in green) in a looping fashion for comparison. All other cells express membrane dtTomato (grey). Lateral orientation with anterior to the left (scale bar: 50 μm).
DOI: https://doi.org/10.7554/eLife.42434.009

## Results

### Mesoderm migration mode and cell shape differ in embryonic versus extra-embryonic regions

The T box transcription factor *Brachyury* is expressed in posterior epiblast cells that form the primitive streak, maintained in cells that delaminate through the streak, then down-regulated once cells progress anteriorly in the mesodermal wings (*Wilkinson et al., 1990*). In order to visualize nascent mesoderm, *Brachyury*-Cre (hereafter referred to as *T*-Cre) transgenic animals, in which a construct encoding Cre cDNA fused to the regulatory elements of the *Brachyury* gene directing gene expression in the primitive streak was randomly inserted (*Feller et al., 2008*; *Stott et al., 1993*), were crossed to a membrane reporter line: Rosa26::membrane dtTomato/membrane GFP (*Muzumdar et al., 2007*) (referred to as mTmG) (*Figure 1*). In *T*-Cre; mTmG embryos, primitive streak and mesoderm-derived cells have green membranes (mG), whereas all other cells have red membranes (mT). Embryos dissected at E6.75 or E7.25 were staged according to *Downs and Davies (1993)* (*Figure 1—figure supplement 1a*) and examined in different orientations by confocal or two-photon excitation live imaging for 8 to 12 hr (*Figure 1*, *Figure 1—figure supplement 1c and d*, *Videos 1* and *2*). Conversion of mT to mG was first observed at Early/Mid Streak (E/MS) stage, and was initially mosaic, which facilitated the tracking of individual migrating cells with high cell shape resolution. From Mid/Late Streak (M/LS) onwards, most primitive streak cells underwent red to green conversion (*Figure 1—figure supplement 1e,f*).

The shape of mesoderm cells and their tracks were recorded through imaging of embryos from different perspectives between ES and Early Bud (EB) stages of development, in order to obtain

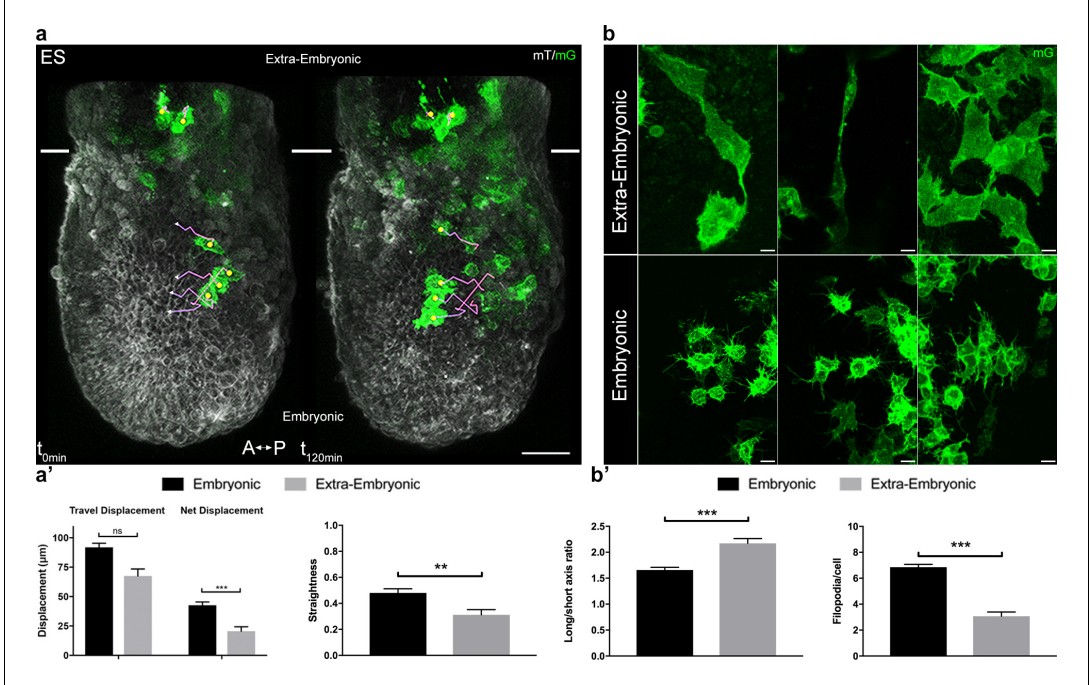

**Figure 2.** Embryonic and extra-embryonic mesoderm populations have different morphology and migration pattern. (a) Z-projections of confocal stacks from a *T*-cre; mTmG embryo dissected at E6.75 (Early Streak), with cell migration tracking for 120 min. Anterior to the left. White lines mark the embryonic/extra-embryonic boundary. (Scale bar: 50 μm). (a') Quantification (mean ± SEM) of travel and net displacement (Left) and path straightness (Right, on a scale of 0 to 1) of embryonic (black, n=34 from 4 Early/Mid Streak embryos) and extra-embryonic (grey, n=17 cells) mesoderm cells. Data can be found in *Table 1* and *Figure 2—source data 1*. (b) Embryonic and extra-embryonic mesoderm cell shapes (images extracted from 4 Late Streak embryos) (Z-projections of two-photon stacks, scale bar: 10 μm). (b') Left: Long/short axis ratio of 2D inner ellipse as quantification of cell stretch (mean ± SEM, n=85 embryonic cells in black, n=83 extra-embryonic cells in grey, out of 8 Mid Streak to Zero Bud stages embryos). Right: Quantification (mean ± SEM) of number of filopodia per cell per time point, in embryonic (black, n=167 cells out of 5 Mid Streak to Early Bud stages embryos) and extra-embryonic (grey, n=28 cells) mesoderm cells. Data can be found in *Table 2*, *Figure 2—source data 2* and *3*. P values were calculated using the Mann–Whitney–Wilcoxon. mG: membrane GFP, in green; mT: membrane dtTomato, in grey.
DOI: https://doi.org/10.7554/eLife.42434.011

The following source data is available for figure 2:

**Source data 1.** Embryonic and extra-embryonic mesoderm cells tracking: List detailing individual cells tracking, volume and surface measurement results.
DOI: https://doi.org/10.7554/eLife.42434.012
**Source data 2.** Embryonic and extra-embryonic mesoderm shape measurements.
DOI: https://doi.org/10.7554/eLife.42434.013
**Source data 3.** Embryonic and extra-embryonic mesoderm cells filopodia: Filopodia number/cell/time point and filopodia length measurements.
DOI: https://doi.org/10.7554/eLife.42434.014

**Table 1.** Tracking details for embryonic and extra-embryonic mesoderm.
Cells were tracked for approximately 150 min. P values were calculated using the Mann–Whitney–Wilcoxon. Data can be found in *Figure 2—source data 1*.

| | Net displacement (μm) | | Travel displacement (μm) | | Straightness | | Mean speed (μm/min) | | |
|---|---|---|---|---|---|---|---|---|---|
| | Mean | SEM | Mean | SEM | Mean | SEM | Mean | SEM | N |
| Extra-embryonic | 20.58 | 3.74 | 67.50 | 6.01 | 0.31 | 0.04 | 0.44 | 0.03 | 17 |
| Embryonic | 42.64 | 2.79 | 91.86 | 3.51 | 0.48 | 0.03 | 0.67 | 0.03 | 34 |
| P-value | 7.93E-05 | | 5.29E-01 | | 1.54E-03 | | 1.99E-05 | | |

DOI: https://doi.org/10.7554/eLife.42434.010

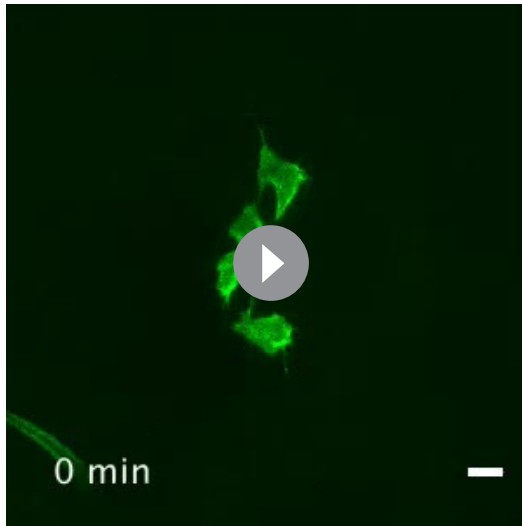

**Video 5.** Extra-embryonic mesoderm migration is characterized by low net displacement. Z-projection of confocal stack from a *T*-Cre; mTmG embryo dissected at E6.75 (Early Streak stage) and imaged for 860 min cropped to show extra-embryonic mesoderm cells. Mesoderm cells express membrane GFP (green) (scale bar: 10 μm).
DOI: https://doi.org/10.7554/eLife.42434.015

images of optimal quality for each embryo region (*Figure 1* and *Videos 1* and *2*). Posterior views (*Figure 1a*) showed proximal to distal primitive streak extension and basal rounding of bottle-shaped cells exiting the streak, as previously described (*Williams et al., 2012*). Lateral views (*Figure 1b*) allowed comparing cells as they migrated laterally in mesodermal wings, or proximally in extra-embryonic region. Anterior views (*Figure 1c*) showed cell movement towards the midline. The imaging time frame did not allow following individual cells from their exit at the primitive streak to their final destination. However, the trajectories we acquired (*Videos 3* and *4*) fitted with the fate maps built using cellular labeling or transplantation (*Kinder et al., 1999*; *Kinder et al., 2001*), or adaptive light sheet microscopy (*McDole et al., 2018*). The first converted (GFP positive) cells in ES embryos dissected around E6.75 usually left the posterior site of the primitive streak to migrate towards the extra-embryonic compartment. Embryonic migration started almost simultaneously, and migration towards both regions proceeded continuously.

Strikingly, migration behavior (*Figure 2a*, *Videos 3* and *4*) and cell shape (*Figure 2b*) were different depending on the region cells migrated into. In the embryonic region, mesoderm cells had a global posterior to anterior path, even though they zigzagged in all directions (proximal-distal, left-right, and even anterior-posterior). Cells did not migrate continuously, but showed alternations of tumbling behavior with straighter displacement, as described for zebrafish mesendoderm progenitors (*Diz-Muñoz et al., 2016*). Embryonic mesoderm cells from ES/MS embryos tracked for 2.5 hr moved at a mean speed of 0.65 μm/min to cover approximately 90 μm and travel a net distance of 40 μm (*Figure 2a'*, *Table 1*). Straightness (the ratio of net over travel displacement, so that a value of 1 represents a linear path) was approximately 0.5 (*Figure 2a'*, *Table 1*). Cells in the extra-embryonic region moved slightly slower (0.45 μm/min) to do approximately 70 μm, but their net displacement (20 μm) and straightness (0.3) were significantly smaller, reflecting trajectories with no obvious directionality (*Figure 2a'*, *Table 1*, and *Video 5*).

Extra-embryonic cells were twice larger in volume, and more elongated (*Figure 2b,b'*, *Table 2*). The increased size can be attributed to a lower frequency of division (*Figure 1—figure supplement 1b*). They had few large protrusions, and filopodia were scarce and short (*Figure 2b,b'*, *Table 2*).

**Table 2.** Cell shape, size, and filopodia comparison between embryonic and extra-embryonic mesoderm.
P values were calculated using the Mann–Whitney–Wilcoxon for surface, long/short axis and filopodia/cell/time point and the t test for volume, and filopodia length. Data can be found in *Figure 2—source data 2* and *3*.

| | Volume (μm³) | | Surface (μm²) | | Long/short axis | | | Filopodia/cell | | Filopodia length (μm) | | |
|---|---|---|---|---|---|---|---|---|---|---|---|---|
| | Mean | SEM | Mean | SEM | Mean | SEM | N | Mean | SEM | Mean | SEM | N |
| Extra-embryonic | 4253.03 | 234.80 | 2438.82 | 105.59 | 2.17 | 0.09 | 83 | 3.07 | 0.32 | 6.20 | 0.37 | 28 |
| Embryonic | 2002.08 | 81.36 | 1308.63 | 39.14 | 1.66 | 0.05 | 85 | 6.86 | 0.21 | 8.00 | 0.14 | 167 |
| P-value | 2.10E-16 | | 5.41E-19 | | 3.04E-05 | | | 3.28E-11 | | 5.24E-04 | | |

DOI: https://doi.org/10.7554/eLife.42434.016

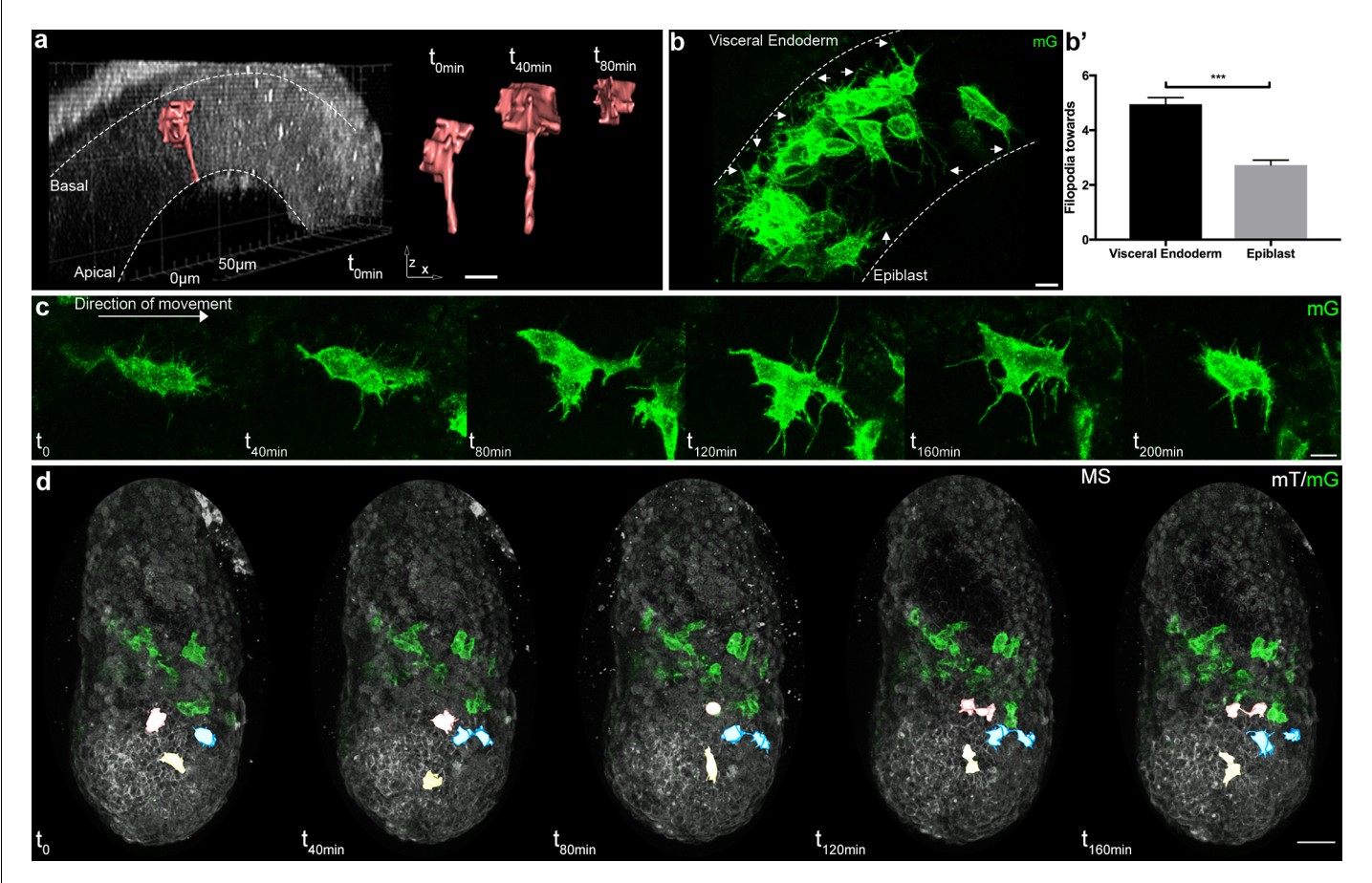

**Figure 3.** Cell shape changes of migrating mesoderm. (a) Cell shape progression of nascent mesoderm delaminating at the primitive streak of a Mid Streak embryo (Z-projection of two-photon stack, scale bar: 10 µm). (b) Mesoderm cells extend filopodia (arrows) towards epiblast and visceral endoderm. Embryo is at Late Streak stage. (Z-projection of two-photon stack, scale bar: 10 µm). (b') Quantification of filopodia per cell per time point as mean ± SEM, n=40 cells out of 4 Late Streak embryos for each, p<0.0001. P value was calculated using the t test. Data can be found in *Figure 3—source data 1*. (c) Montage of a mesoderm cell (from a Mid Streak stage embryo) displaying seeking behavior (Z-projection of two-photon stack, scale bar: 10 µm). (d) Mesoderm cells are highlighted, through manual segmentation, in red, blue and yellow to track cell behavior after division (Z-projection of confocal stacks from a Mid Streak stage embryo, anterior view, scale bar: 50 µm). mG: membrane GFP, in green; mT: membrane dtTomato, in grey.
DOI: https://doi.org/10.7554/eLife.42434.017

The following source data is available for figure 3:

**Source data 1.** Mesoderm cells filopodia extended towards Visceral Endoderm and Epiblast.
DOI: https://doi.org/10.7554/eLife.42434.018
**Source data 2.** Quantification of daughter cells trajectory.
DOI: https://doi.org/10.7554/eLife.42434.019
**Source data 3.** Quantification of trajectory of cells in close proximity.
DOI: https://doi.org/10.7554/eLife.42434.020
**Source data 4.** Quantification of trajectory of cells after collision.
DOI: https://doi.org/10.7554/eLife.42434.021

## Mesoderm cells have distinct morphology depending on their interaction with different germ layers

Cells passing through the primitive streak were, as reported (*Ramkumar et al., 2016*; *Williams et al., 2012*), bottle shaped with a basal round cell body and an apical thin projection (*Figure 3a*). Mesoderm cells in contact with the epiblast and visceral endoderm sent thin protrusions towards their respective basal membranes (*Figure 3b,b'*, and *Video 6*). Interestingly, the density of thin protrusions was much higher in cells in contact with the visceral endoderm. As this phenotype

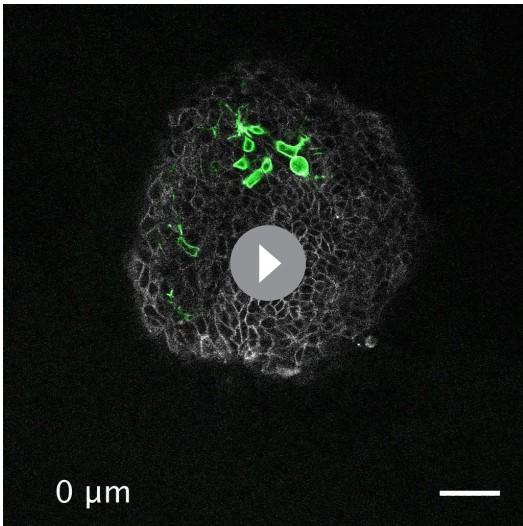

**Video 6.** Mesoderm extends filopodia towards epiblast and visceral endoderm. Two-Photon stack of a *T*-Cre; mTmG embryo at Late Streak stage. The stack progresses from anterior to posterior. Mesoderm cells express membrane GFP (green); all other cells express membrane dtTomato (grey) (scale bar: 50 μm).
DOI: https://doi.org/10.7554/eLife.42434.022

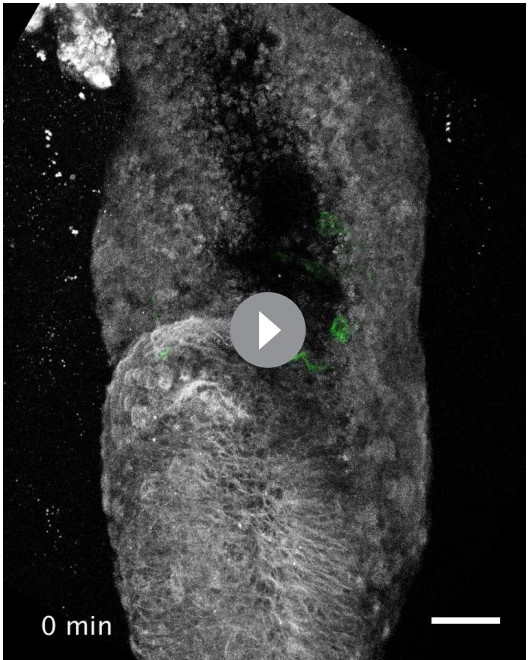

**Video 7.** Searching behavior. Z-projection of two-photon stack from a *T*-Cre; mTmG embryo at Mid Streak stage imaged for 100 min. Arrow points at embryonic mesoderm cell. Mesoderm cells express membrane GFP (green); all other cells express membrane dtTomato (grey) (scale bar: 50 μm).
DOI: https://doi.org/10.7554/eLife.42434.023

was observed as early as ES in the posterior region, it could reflect the putative signaling role of the proximal posterior visceral endoderm, which remains coherent along gastrulation (*Kwon et al., 2008*). Intercalation of prospective definitive endoderm cells in the visceral endoderm layer, which occurs from LS stage onwards (*Viotti et al., 2014*), was very rarely observed, either because Brachyury does not label endoderm progenitors (*Burtscher and Lickert, 2009*), or because the *T*-Cre transgenic line does not bear the regulatory elements driving T expression in prospective endoderm. Cells in tight clusters surrounded by other mesoderm cells in the wings had a smoother contour, with the caveat that protrusions couldn't be visualized between cells of similar membrane color. Reconstruction of cells in the anterior part of the wings, where recombination was incomplete, showed thin protrusions between mesoderm cells (*Figure 1c*). Cells also extended long broad projections, which spanned several cell diameters and were sent in multiple directions before translocation of the cell body, in what seemed a trial and error process (*Figure 3c* and *Videos 1*, *2* and *7*). The presence of potential leader cells could not be assessed, as the first cells converted to green are not the most anterior ones (*Figure 1—figure supplement 1e*). Nonetheless, cells with scanning behavior were observed at all times, which suggests that all cells are capable to explore their surroundings.

## Impact of cell-cell contact on trajectory within the mesoderm layer

To address the collectiveness of mesoderm migration, we assessed the impact of cell proximity on behavior by comparing the trajectories of daughter cells after mitosis, pairs of unrelated cells that were in immediate proximity at the beginning of observation, and cells colliding along the way.

As expected from fate mapping experiments, daughter cells resulting from mitosis within the mesoderm layer followed close and parallel trajectories (*Figure 3d* and *Figure 3—source data 2*). They travelled a similar net distance over 204 min (net displacement ratio: 0.91 ± 0.01, n=12 pairs from four embryos at E/MS stage), in the same direction (angle: 7 ± 1.13°), with one daughter cell displaying a higher straightness (travel displacement ratio: 0.61 ± 0.07). They remained close to one another (mean distance between daughter cells: 15.6 ± 2.35 μm), but not directly apposed.

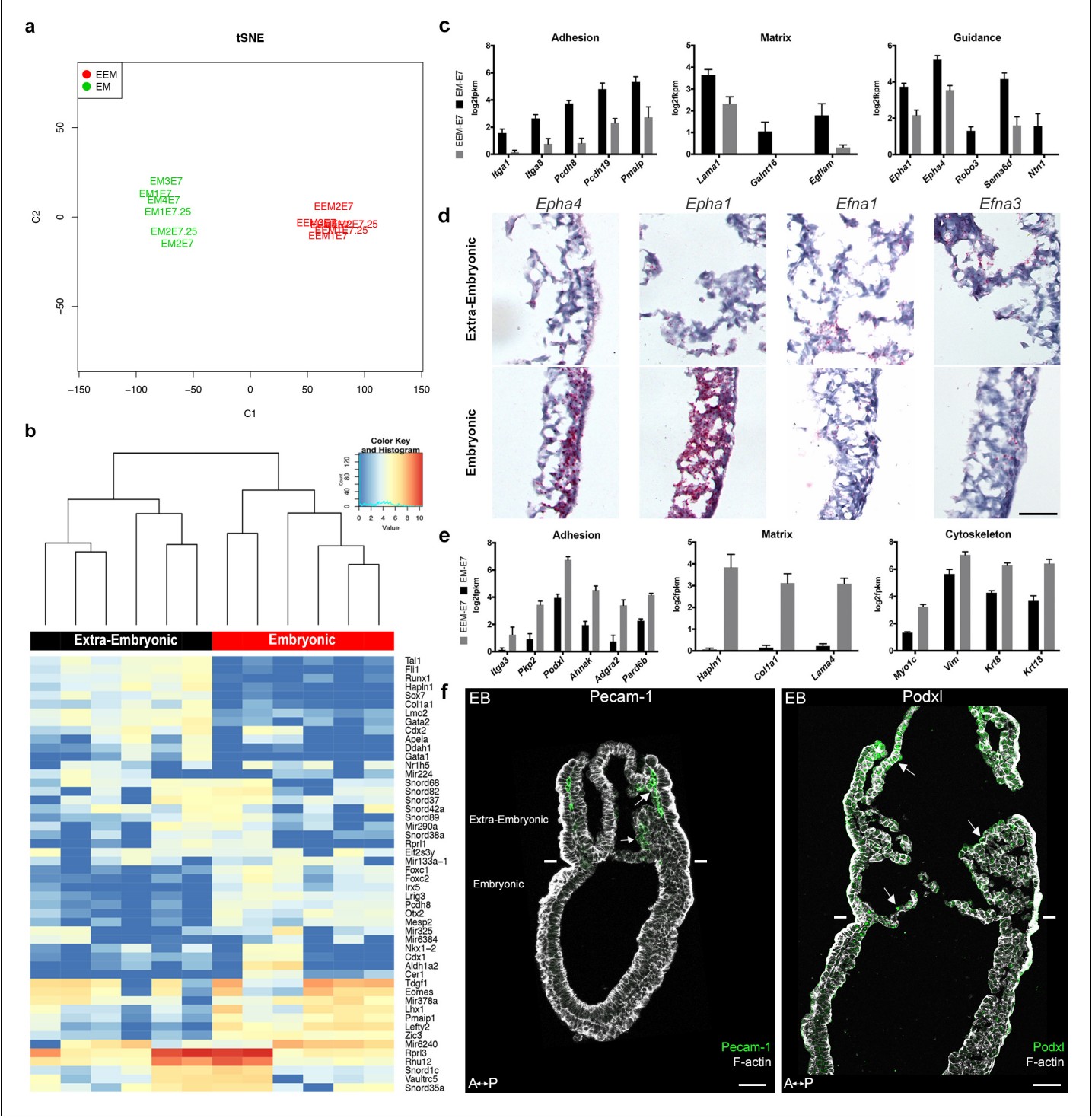

**Figure 4.** Transcriptomes of mesoderm populations identify differences between embryonic and extra-embryonic mesoderm. (**a**) t-Distributed Stochastic Neighbor Embedding (t-SNE) confirms grouping of similar biological samples at E7 (Mid Streak stage) and E7.25 (Late Streak stage). EM: Embryonic mesoderm; EEM: Extra-Embryonic mesoderm. More sample information can be found in *Figure 4—source data 1*. (**b**) Heat map showing differentially expressed genes between embryonic and extra-embryonic mesoderm with the highest statistical significance. (**c, e**) Selection of genes with higher expression in embryonic (**c**) or extra-embryonic (**e**) mesoderm, represented as mean ± SEM of log2 fpkm at E7 (n=4 biological replicates, p<0.01), with embryonic in black and extra-embryonic in grey. All represented genes are also significantly differentially regulated at E7.25. Data can be found in *Figure 4—source data 2* and *3*. (**d**) In situ hybridization of sagittal sections (anterior to the left) from Zero to Early Bud stages embryos highlighting transcripts for *Epha4*, *Epha1*, *Efna1* and *Efna3*, represented by red dots, in the posterior region. Entire embryo sections are shown in *Figure 4—figure supplement 1*. (Scale bar: 100 μm) (**f**) Sagittal sections (anterior to the left) from Early Bud stage embryos stained for Platelet

*Figure 4 continued on next page*

*Figure 4 continued*
endothelial cell adhesion molecule 1 (left panel, Pecam-1 in green, Z-projection of confocal stack), Podocalyxin (right panel, Podxl in green, optical slice), and F-actin (Phalloidin, grey). See same section stained for mGFP in *Figure 4—figure supplement 1*. White lines mark the embryonic/extra-embryonic boundary. (Scale bars: 50 µm).
DOI: https://doi.org/10.7554/eLife.42434.024
The following source data and figure supplement are available for figure 4:

**Source data 1.** Description and quality control of samples used for RNA-seq.
DOI: https://doi.org/10.7554/eLife.42434.026
**Source data 2.** Expression Levels.
DOI: https://doi.org/10.7554/eLife.42434.027
**Source data 3.** Ranked list of differential expression.
DOI: https://doi.org/10.7554/eLife.42434.028
**Figure supplement 1.** Mouse embryonic and extra-embryonic mesoderm transcriptomes.
DOI: https://doi.org/10.7554/eLife.42434.025

Interestingly, they stayed linked by thin projections for hours, even when separated by other cells (*Figure 3d*).

Pairs of cells that were in immediate proximity at time zero traveled a similar distance over 152 min (net displacement ratio: 0.88 ± 0.03, n=13 pairs from four embryos), in the same direction (angle: 9.9° ± 2.77); they remained close to one another (travel displacement ratio and final distance were respectively 0.75 ± 0.04 and 15.2 µm ± 3.84 µm) (*Figure 3—source data 3*). Contact between cell protrusions could be spotted in most pairs.

Mesoderm cell migration is often compared to neural crest migration, as both cell types arise through epithelial-mesenchymal transition (*Roycroft and Mayor, 2016*). An important feature of neural crest migration is contact inhibition of locomotion, where cells that collide tend to move in opposite directions. In contrast, most mesoderm cells stayed in contact upon collision (*Figure 3—source data 4*): 16 out of 24 cell pairs from 5 ES to Zero Bud (0B) embryos remained attached for 2.5 hr (one briefly lost contact before re-joining), 3/24 stayed joined for around 1 hr, and 5/24 pairs separated instantly. We segmented 8 pairs for 166 min, and observed a mean distance at the end of tracking of 52.5 ± 32.5 µm; they followed parallel trajectories (angle: 8.25 ± 1.7°, n=8 pairs), for a similar net distance (net displacement ratio: 0.85 ± 0.07; travel displacement ratio: 0.64 ± 0.11). Thin projections could occasionally be observed between them after contact.

Those data suggest that cells coming in close proximity tend to have a similar behavior. The presence of thin projections between them may reflect cell-to-cell communication.

## Embryonic and extra-embryonic mesoderm molecular signatures

Embryonic and extra-embryonic mesoderm cells were isolated through fluorescence (GFP)-assisted cell sorting from E7.5 MS and LS *T*-Cre; mT/mG embryos in order to generate transcriptomes. Biological replicates (4 MS, 2 LS) from both stages resulted in grouping of samples according to the embryonic region (*Figure 4a,b*). Non-supervised clustering based on embryonic and extra-embryonic gene signatures identified by single cell sequencing in *Scialdone et al. (2016)* showed that samples segregated as expected (not shown). We performed pairwise comparison of embryonic and extra-embryonic samples obtained at both stages and selected the genes that were consistently differentially expressed with a fold change >2. Gene ontology analysis of genes clusters enriched either in embryonic or extra-embryonic mesoderm highlighted expected developmental (angiogenesis and hematopoiesis in extra-embryonic, somitogenesis in embryonic), and signaling (BMP and VEGF in extra-embryonic, Wnt and Notch in embryonic) biological processes (*Figure 4—figure supplement 1a,a'*). Interestingly, differences were also seen in gene clusters involved in migration, adhesion, cytoskeleton, and extracellular matrix organization.

Genes with known expression pattern in gastrulation embryos found enriched in embryonic mesoderm included well-described transcription factors, as well as FGF, Wnt, Notch, TGFβ and Retinoic Acid pathways effectors (*Figure 4—figure supplement 1b*). Genes expected to be more expressed in extra-embryonic mesoderm included the transcription factors *Ets1* and *Tbx20*, and several members of the TGFβ pathway (*Inman and Downs, 2007*; *Pereira et al., 2011*) (*Figure 4—figure supplement 1c*). Primitive hematopoiesis, the initial wave of blood cell production which gives rise to

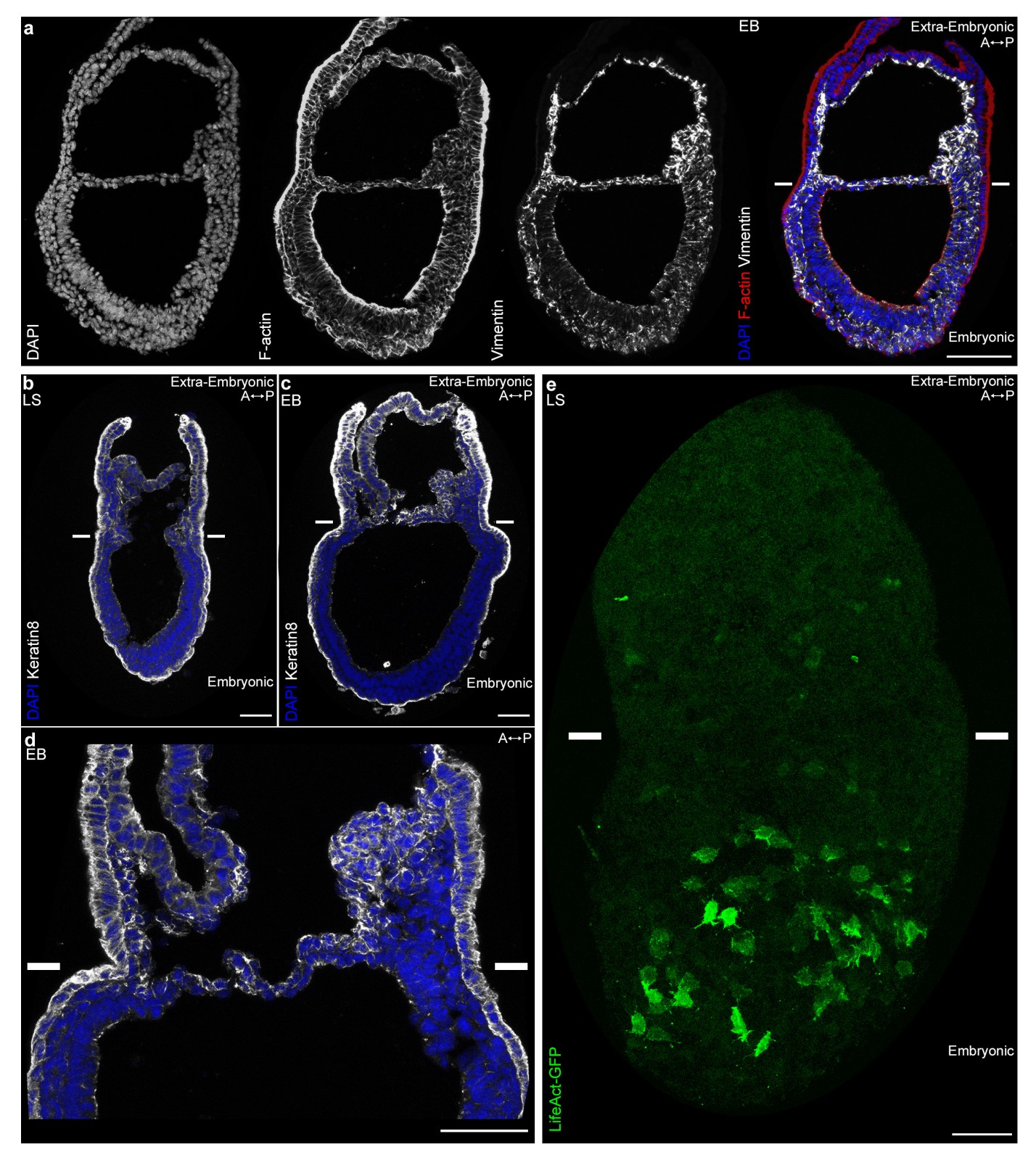

**Figure 5.** Embryonic and extra-embryonic mesoderm cells have distinct cytoskeleton composition. (**a**) Z-projections of confocal stack of a sagittal section from an Early Bud stage embryo stained for Vimentin, F-Actin (Phalloidin), and nuclei (DAPI). (**b, d**) Z-projections of confocal stacks of sagittal sections from Late Streak (**b**) and Early Bud (c: 20x, d: 40x) stages embryos stained for Keratin 8 (grey) and nuclei (DAPI, blue). (**e**) Z-projection of two-photon stack of a whole-mount *T*-Cre; LifeAct-GFP (green) Late Streak embryo. Anterior is to the left. (Scale bars: 50 μm).

DOI: https://doi.org/10.7554/eLife.42434.029

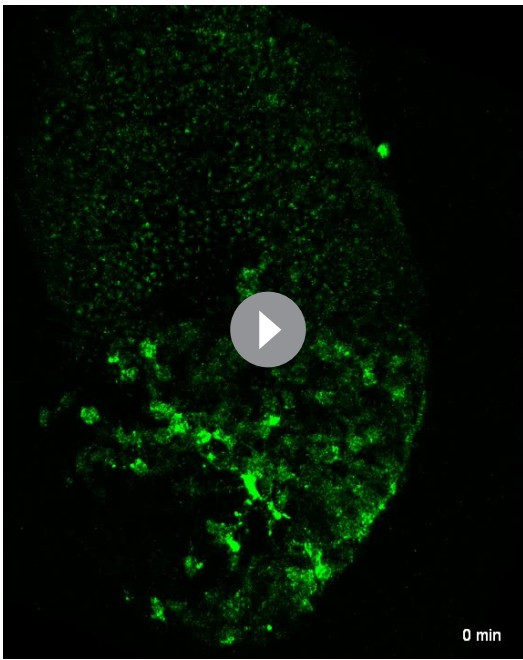

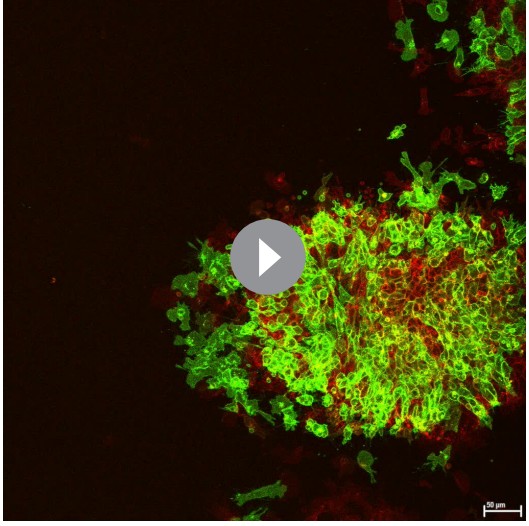

**Video 9.** Mesoderm explant. Z-projection of confocal stack of mesoderm explant from a *T*-Cre; mTmG embryo dissected at E7.5 (Late Streak stage). The explant was imaged for 750 min every 15 min.
DOI: https://doi.org/10.7554/eLife.42434.031

**Video 8.** LifeAct-GFP expression is higher in embryonic mesoderm. 3D snapshots of two-photon stacks from a *T*-Cre; LifeAct-GFP embryo dissected at E7.25 (Late Streak stage) and imaged for 295 min. Images were processed with the ZEN blue denoise function. LifeAct-GFP (in green) highlights F-actin. The bright specks in extra-embryonic on the right side are debris. Lateral orientation with anterior to the left (3D scale bar: 50 μm).
DOI: https://doi.org/10.7554/eLife.42434.030

primitive erythrocytes, macrophages, and mega-karyocytes, takes place around E7.25 in hemogenic angioblasts of the blood islands (*Lacaud and Kouskoff, 2017*). Expression of genes involved in hemangioblast development, endothelium differentiation, and hematopoiesis increased from MS to LS (*Figure 4f* and *Figure 4—figure supplement 1c*). In addition, we confirmed two extra-embryonic genes identified through subtractive hybridization at E7.5 (*Kingsley et al., 2001*): *Ahnak* (*Figure 4—figure supplement 1c*), see also *Downs et al. (2002)* and the imprinted gene *H19* (*Figure 4—figure supplement 1c*).

Of particular interest among the genes with higher expression in embryonic mesoderm for which no expression data was available at the stage of development were genes related to matrix (*Lama1, Galnt16, Egflam*), adhesion (*Itga1, Itga8, Pcdh8, Pcdh19, Pmaip1*), and guidance (*Epha1 and 4, Robo3, Sema6d, Ntn1*) (*Figure 4c*). *Epha4* expression in the mouse embryo has been described in the trunk mesoderm and developing hindbrain at Neural Plate (NP) stage (*Nieto et al., 1992*). In LS embryos, *Epha4* expression was higher in the primitive streak and embryonic mesoderm (*Figure 4d* and *Figure 4—figure supplement 1e*). Dynamic *Epha1, Efna1 and Efna3* expression patterns have been shown during gastrulation (*Duffy et al., 2006*). In LS/0B embryos, *Epha1* mRNA was present in the primitive streak, mostly in its distal part. Its ligand *Efna1* was in the primitive streak with an inverse gradient, and was mainly expressed in the extra-embryonic region, notably in amnion and in chorion. *Efna3* was very abundant in the chorion (*Figure 4d* and *Figure 4—figure supplement 1e*).

In parallel, in extra-embryonic mesoderm, we found higher expression of distinct sets of genes with putative roles in guidance (*Unc5c, Dlk1, Scube2, Fzd4*), matrix composition (*Hapln1, Col1a1, Lama4*), adhesion (*Itga3, Pkp2, Podxl, Ahnak, Adgra2, Pard6b*), Rho GTPase regulation (*Rasip1, Stard8, Rhoj*), and cytoskeleton (*Myo1c, Vim, Krt8 and Krt18*) (*Figure 4e* and not shown). Interestingly, Podocalyxin (*Podxl*) was abundant in extra-embryonic mesoderm (*Figure 4f* and *Figure 4—figure supplement 1d*), which fits with data from embryo and embryoid body single cell sequencing showing that *Podxl* is a marker for early extra-embryonic mesoderm and primitive erythroid progenitors of the yolk sac (*Zhang et al., 2014*).

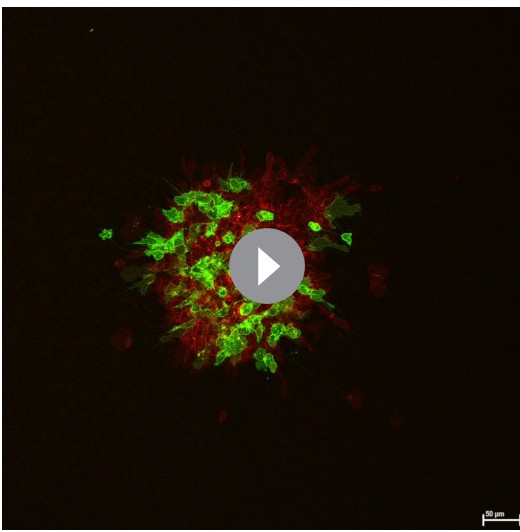

**Video 10.** *Rhoa^Δmesoderm* explant undergoes compaction before cell migration. Z-projection of confocal stack of mesoderm explant from a *T*-Cre; mTmG; *Rhoa* fl/- embryo dissected at E7.5 (Late Streak stage). The explant was imaged for 750 min every 15 min.
DOI: https://doi.org/10.7554/eLife.42434.032

## Embryonic and extra-embryonic mesoderm cells have distinct cytoskeletal composition

In view of the differences in cell shape and migration, we focused on the cytoskeleton, in particular actin, and intermediate filaments proteins (Vimentin and Keratins). Vimentin was found in all mesoderm cells as expected, but more abundant in extra-embryonic mesoderm (*Figure 5a*). Remarkably, within the mesoderm layer, Keratin 8 was selectively expressed in extra-embryonic mesoderm cells (amniochorionic fold, amnion, chorion, and developing allantois) (*Figure 5b–d*). In contrast, the filamentous actin (F-actin) network, visualized by Phalloidin staining, seemed denser in embryonic mesoderm (*Figure 5a*). To visualize F-actin only in mesoderm, we took advantage of a conditional mouse model expressing Lifeact-GFP, a peptide that binds specifically to F-actin with low affinity, and thus reports actin dynamics without disrupting them (*Schachtner et al., 2012*). Live imaging of *T*-Cre; LifeAct-GFP embryos at MS and LS stage confirmed that while LifeAct-GFP positive filaments could be visualized clearly in embryonic mesoderm cells, GFP was weaker and diffuse in extra-embryonic mesoderm (*Figure 5e* and *Video 8*).

## Extra-embryonic mesoderm migration is Rho GTPases independent

Rho GTPases are molecular switches that relay signals from cell surface receptors to intracellular effectors, leading to a change in cell behavior (*Hodge and Ridley, 2016*). They are major regulators of cytoskeletal rearrangements (*Hall, 1998*), and the spatiotemporal fine regulation of Rho GTPases activities determines cytoskeletal dynamics at the subcellular level (*Spiering and Hodgson, 2011*). Therefore, inactivation of a given Rho GTPase may result in variable consequences depending on cell type and context. We previously established that *Sox2*-Cre mediated deletion of *Rac1* in the epiblast before onset of gastrulation causes impaired migration of embryonic mesoderm while extra-embryonic mesoderm migration is less severely affected (*Migeotte et al., 2011*). We thus hypothesized that Rho GTPases might be differentially regulated in cells invading both regions, resulting in some of the observed distinctions in cytoskeletal dynamics, cell shape and displacement mode.

Mutations were induced in cells transiting the primitive streak by crossing heterozygous wild-type/null *Rhoa* (*Jackson et al., 2011*) or *Rac1* (*Walmsley et al., 2003*) animals bearing the *T*-Cre transgene with animals homozygous for their respective conditional alleles bearing the mTmG reporter (mutant embryos are referred to as *Rhoa^Δmesoderm* and *Rac1^Δmesoderm*). The phenotypes of *Rhoa^Δmesoderm* and *Rac1^Δmesoderm* embryos were less severe than that of *Rhoa^Δepi* and *Rac1^Δepi* embryos (our unpublished data and *Migeotte et al., 2011*) (*Figure 6—figure supplement 1*). Mutants were morphologically indistinguishable at E7.5. At E8.5, *Rhoa^Δmesoderm* embryos were identified, though with incomplete penetrance, as being slightly smaller than their wild-type littermates (11/12 mutants had a subtle phenotype, including five with reduced numbers of somites, and six with abnormal heart morphology) (*Figure 6—figure supplement 1a*). By E9.5, all *Rhoa^Δmesoderm* embryos had an obvious phenotype (12/12 mutants had a small heart, 9/12 had a reduced number of somites, 2/12 had an open neural tube, 2/12 had a non-fused allantois) (*Figure 6—figure supplement 1b*). *Rac1^Δmesoderm* embryos also had subtle phenotypes at E8.5 (15/16 embryos were slightly smaller than wild-type littermates, 4/16 had a small heart) (*Figure 6—figure supplement 1d*). In situ hybridization for *Brachyury* showed weaker staining in the tail region in 5/10 mutant embryos, indicative of reduced presomitic mesoderm (*Figure 6—figure supplement 1c*). By E9.5, all mutants had

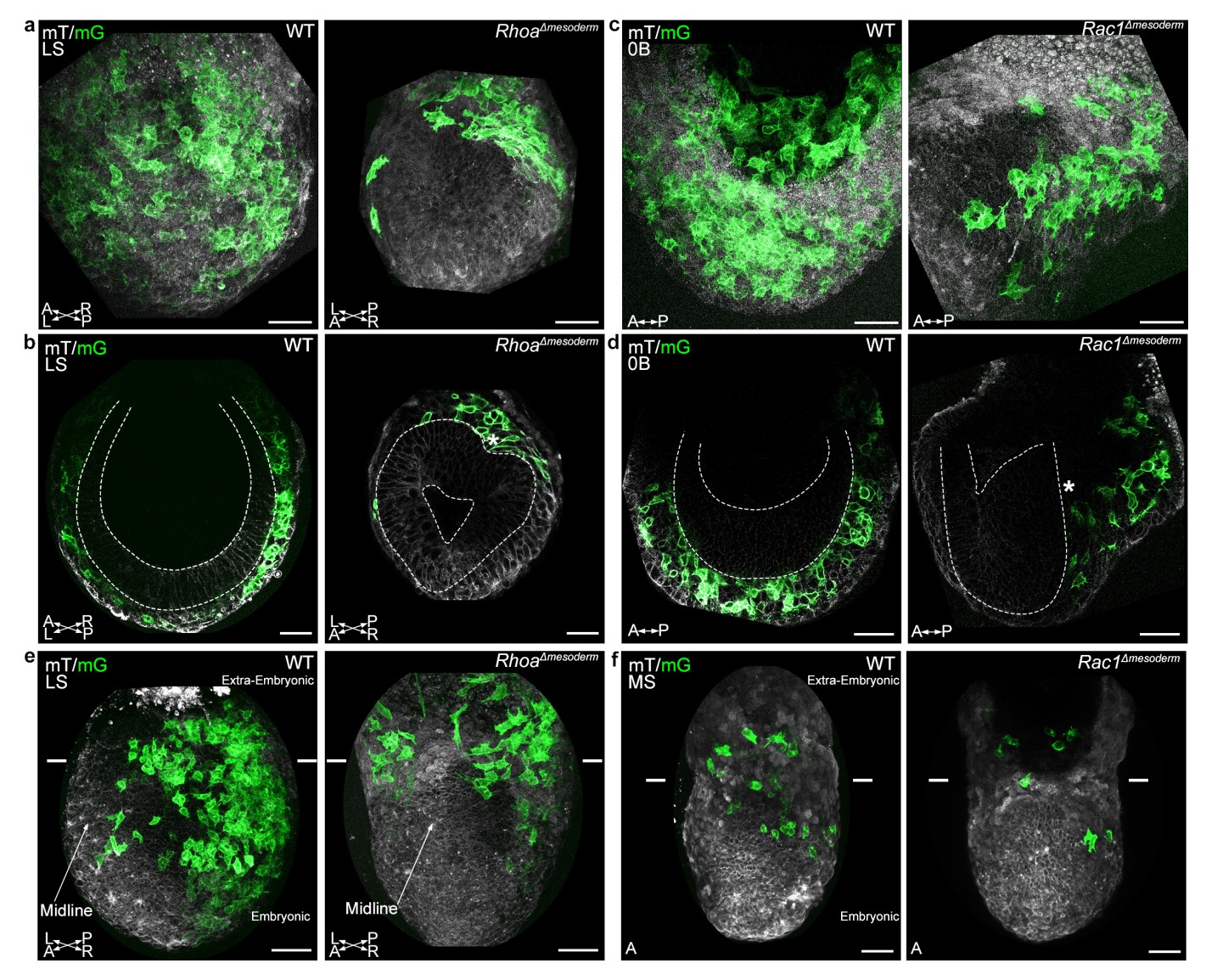

**Figure 6.** *Rhoa* and *Rac1* mesoderm-specific mutants display impaired migration of embryonic mesoderm. (**a**) Z-projection and (**b**) frontal optical slice of wild-type (WT) and *Rhoa^Δmesoderm* embryos. Dashed lines mark the epiblast. (**c**) Z-projection and (**d**) sagittal optical slice of wild-type and *Rac1^Δmesoderm* embryos highlighting the accumulation of mesoderm cells next to the primitive streak (*). Dashed lines mark the epiblast. (**e**) Z-projections of wild-type and *Rhoa^Δmesoderm* embryos (oblique anterior view, posterior to the right, two-photon) show impaired mesoderm migration in embryonic but not extra-embryonic regions. (**f**) Z-projections of wild-type and *Rac1^Δmesoderm* embryos (anterior view, confocal) show impaired mesoderm migration in embryonic but not extra-embryonic regions. Each mutant is compared to a wild-type littermate in similar orientation. White lines mark the embryonic/extra-embryonic boundary. mG: membrane GFP, in green; mT: membrane dtTomato, in grey. (Scale bar: 50 μm).

DOI: https://doi.org/10.7554/eLife.42434.033

The following source data and figure supplements are available for figure 6:

**Figure supplement 1.** Phenotypes of *Rac1* and *Rhoa* mesoderm-specific mutants post gastrulation.
DOI: https://doi.org/10.7554/eLife.42434.034

**Figure supplement 1—source data 1.** Tracking of embryonic *Rhoa^Δmesoderm* and *Rac1^Δmesoderm* mesoderm cells.
DOI: https://doi.org/10.7554/eLife.42434.035

**Figure supplement 1—source data 2.** Tracking of extra-embryonic *Rhoa^Δmesoderm* and *Rac1^Δmesoderm* mesoderm cells.
DOI: https://doi.org/10.7554/eLife.42434.036

**Figure supplement 2.** Cellular details in *Rhoa* and *Rac1* mesoderm-deleted embryos.
DOI: https://doi.org/10.7554/eLife.42434.037

**Figure supplement 2—source data 1.** Quantification of cell shape in mesoderm explants from wild-type and *Rhoa^Δmesoderm* embryos.

*Figure 6 continued on next page*

*Figure 6 continued*

DOI: https://doi.org/10.7554/eLife.42434.038

abnormal heart morphology and reduced body length, and 3 embryos out of 9 were severely delayed (*Figure 6—figure supplement 1e*). At E10.5, penetrance was complete; 7/7 embryos had a short dysmorphic body and pericardial edema (not shown). The phenotypic variability at early time points likely reflects mosaicism of *T*-Cre mediated recombination.

Embryonic mesoderm explants from E7.5 MS/LS mTmG; *Rac1*$^{\Delta mesoderm}$ or *Rhoa*$^{\Delta mesoderm}$ embryos were plated on fibronectin. In wild-type explants, cells showed a radial outgrowth from the explants, displaying large lamellipodia in the direction of migration (*Video 9*). After cell-cell contact, they remained connected through long thin filaments. *Rhoa* deficient explants showed less release of individual cells (*Figure 6—figure supplement 2a*). *Rhoa* mutant cells appeared more cohesive, and were rounder than wild-type cells (*Figure 6—figure supplement 2c*). Remarkably, live imaging of explants from *Rhoa* mutant embryos showed a phase of compaction preceding cell migration (*Video 10*; 2/4 *Rhoa*$^{\Delta mesoderm}$ mutant explants displayed compaction). In *Rac1* explants (*Figure 6— figure supplement 2b*), GFP-expressing cells remained within the domain of the dissected explant and displayed pycnotic nuclei, while wild-type non-GFP cells could migrate. Live imaging could not be performed as 4 out of 5 mutant explants detached from the plate. This is similar to explants from *Rac1* epiblast-specific mutants (*Migeotte et al., 2011*), and is attributed to lack of adhesion-dependent survival signals.

Live imaging of mTmG; *Rhoa*$^{\Delta mesoderm}$ or *Rac1*$^{\Delta mesoderm}$ embryos dissected at E6.75 or 7.25 (*Figure 6*) showed that the majority of *Rhoa* and *Rac1* mesoderm-specific mutants (4/8 for *Rhoa*, 6/9 for *Rac1*) displayed an accumulation of cells at the primitive streak, which formed a clump on the posterior side between epiblast

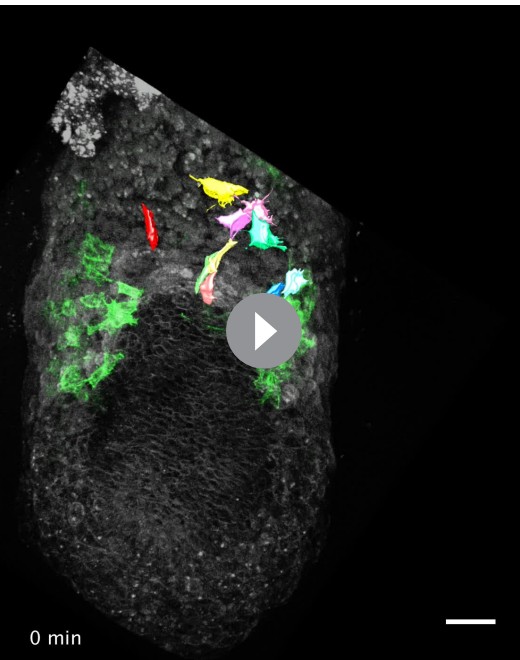

**Video 12.** Mesoderm migration tracking in *Rhoa*$^{\Delta mesoderm}$ embryo. 3D snapshots of stacks from a *T*-Cre; mTmG; *Rhoa* fl/- embryo dissected at E7.25 (Mid Streak stage) and imaged using two-photon microscopy for 120 min showing highlighted cells, which are tracked throughout the time lapse. The video shows the highlighted cells first, then the original images (Membrane GFP, in green) in a looping fashion for comparison. All other cells express membrane dtTomato (grey). Anterior oblique orientation with anterior to the left (scale bar: 50 µm).
DOI: https://doi.org/10.7554/eLife.42434.040

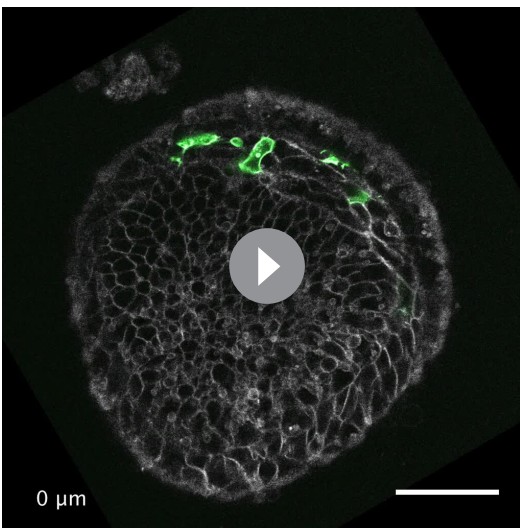

**Video 11.** *Rhoa*$^{\Delta mesoderm}$ embryos display an accumulation of mesoderm near the primitive streak. Two-photon Z stack of a *T*-Cre; mTmG; *Rhoa* fl/- embryo at Late Streak stage. Mesoderm cells express membrane GFP (green); all other cells express membrane dtTomato (grey). Anterior oblique orientation with anterior to the left (scale bar: 50 µm).
DOI: https://doi.org/10.7554/eLife.42434.039

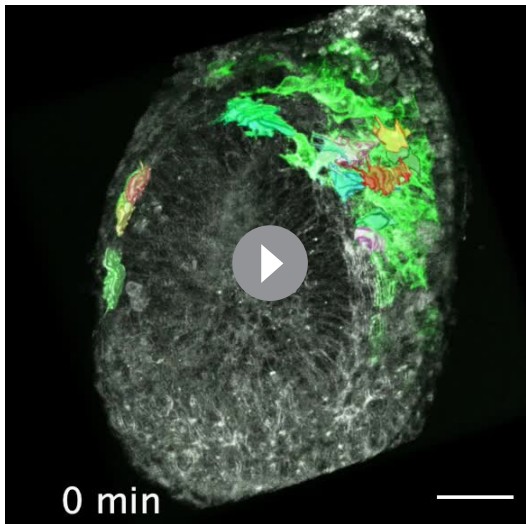

**Video 13.** Mesoderm migration tracking in *Rhoa*<sup>Δmesoderm</sup>embryo. 3D snapshots of stacks from a *T*-Cre; mTmG; *Rhoa* fl/- embryo dissected at E7.25 (Mid Streak stage) and imaged using two-photon microscopy for 100 min showing highlighted cells, which are tracked throughout the time lapse. The video shows the highlighted cells first, then the original images (Membrane GFP, in green) in a looping fashion for comparison. All other cells express membrane dtTomato (grey). Anterior oblique orientation with anterior to the left (scale bar: 50 μm).
DOI: https://doi.org/10.7554/eLife.42434.041

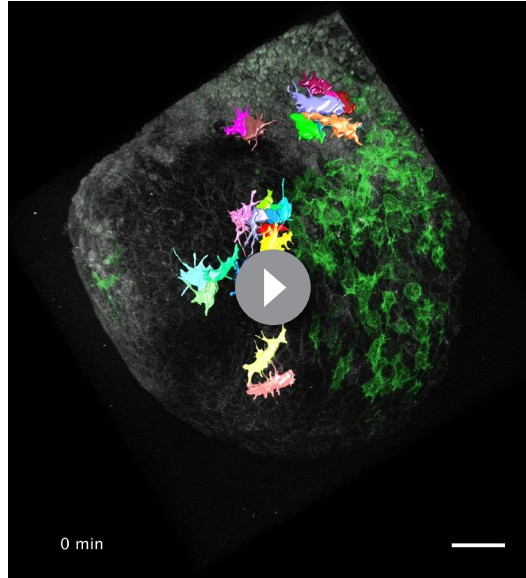

**Video 14.** Mesoderm migration tracking in *Rac1*<sup>Δmesoderm</sup> embryo. 3D snapshots of two-photon stack from a *T*-Cre; mTmG; *Rac1* fl/- embryo dissected at E7.25 (Mid Streak stage) and imaged for 80 min showing highlighted cells, which are tracked throughout the time lapse. The video shows the highlighted cells first, then the original images (Membrane GFP, in green) in a looping fashion for comparison. All other cells express membrane dtTomato (grey). Lateral orientation with anterior to the left (scale bar: 50 μm).
DOI: https://doi.org/10.7554/eLife.42434.042

and visceral endoderm (*Figure 6a–d* and *Video 11*), indicating a mesoderm migration defect. Interestingly, although embryonic mesoderm migration was impaired, with only a handful of cells visible on the anterior side by E7.5, extra-embryonic mesoderm migration was maintained (*Figure 6e,f* and *Videos 12–15*). There was a significant decrease in embryonic, but not extra-embryonic mesoderm cells speed for both *Rhoa* and *Rac1* mutants, compared to wild-type embryos (*Figure 6—figure supplement 1f*). Similarly, in embryos deleted for *Rac1* in epiblast and epiblast-derived cells upon *Sox2*-Cre (*Hayashi et al., 2002*) recombination, GFP positive mesoderm cells were dispersed in the extra-embryonic region, while embryonic mesoderm cells were confined in a bulge adjacent to the primitive streak (*Figure 6—figure supplement 2d*). Accordingly, staining for mesoderm-derived vascular structures (Pecam-1) in the yolk sac at E8.5 showed no difference between mutant and wild-type embryos (*Figure 6—figure supplement 2e,f*). Those findings suggest that extra-embryonic mesoderm cells either do not rely on Rac1 and Rhoa for movement, or are able to compensate for loss of *Rac1* or *Rhoa*, which is consistent with their paucity in actin-rich protrusions.

## Discussion

Mesoderm cell delamination from the epiblast requires basal membrane disruption, apical constriction, loss of apicobasal polarity, changes in intercellular adhesion, and acquisition of motility (*Nieto et al., 2016*). The transcriptional network and signaling pathways involved in epithelial-mesenchymal transition are conserved (*Ramkumar and Anderson, 2011*). However, pre-gastrulation embryo geometry varies widely between species, which has important consequences on interactions between germ layers and mechanical constrains on nascent mesoderm cells (*Williams and Solnica-Krezel, 2017*). Live imaging of mouse embryo has allowed recording posterior epiblast

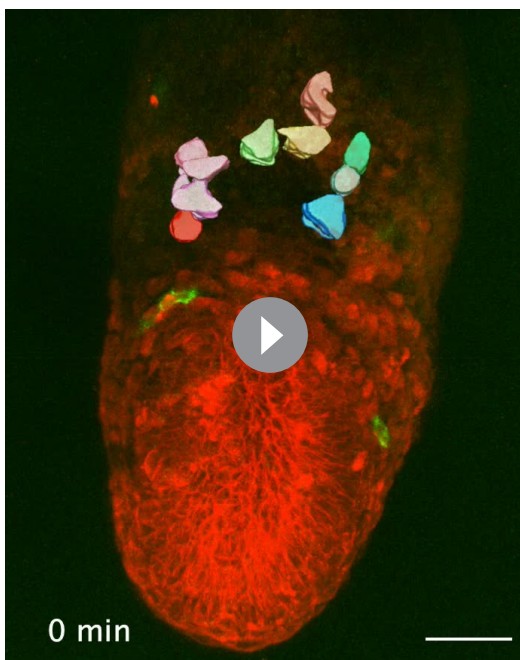

**Video 15.** Mesoderm migration tracking in *Rac1*$^{\Delta mesoderm}$ embryo. 3D snapshots of confocal stacks from a *T-Cre; mTmG; Rac1* fl/- embryo dissected at E6.75 (Early Streak stage) and imaged for 120 min showing highlighted cells, which are tracked throughout the time lapse. The video shows the highlighted cells first, then the original images (Membrane GFP, in green) in a looping fashion for comparison. All other cells express membrane dtTomato (red). Anterior orientation (scale bar: 50 µm).
DOI: https://doi.org/10.7554/eLife.42434.043

rearrangements, as well as cell passage through the primitive streak (*Ramkumar et al., 2016*; *Williams et al., 2012*). Contrary to the chick embryo, there is no global epiblast movement towards the primitive streak in the mouse. However, cell shape changes, including apical constriction and basal rounding, are similar. Morphological data on mouse mesoderm cells acquired through scanning electron microscopy of whole mount samples (*Migeotte et al., 2011*), and transmission electron microscopy of embryo sections (*Spiegelman and Bennett, 1974*) revealed an array of stellate individual cells linked by filopodia containing a lattice of microfilaments.

We took advantage of mosaic labeling of nascent mesoderm to define the dynamics of cell shape changes associated with mesoderm migration. Cells just outside the streak retracted the long apical protrusion and adopted a round shape with numerous filopodia making contacts with adjacent, but also more distant mesoderm cells. In mesodermal wings, cells close to the epiblast were more loosely apposed, and extended fewer filopodia towards its basal membrane, compared to cells adjacent to the visceral endoderm, which were tightly packed and displayed numerous filopodia pointing to the visceral endoderm basal membrane.

Cells travelling in a posterior to anterior direction towards the midline displayed long protrusions, up to twice the cell body size, which extended, retracted, occasionally bifurcated, several times before the cell body itself initiated movement, suggesting an explorative behavior. Remarkably, extension of long protrusions was not limited to the first row of cells. Migration was irregular in time and space, as cells often stopped and tumbled, and displayed meandrous trajectories. After division, cells remained attached by thin protrusions. Contrary to neural crest cells, mesoderm cells did not show contact inhibition of locomotion. Cells in close proximity tended to follow parallel paths.

Extra-embryonic mesoderm first accumulates between extra-embryonic ectoderm and visceral endoderm at the posterior side of the embryo, leading to formation of the amniochorionic fold that bulges into the proamniotic cavity (*Pereira et al., 2011*). This fold expands, and lateral extensions converge at the midline. Accumulation and coalescence of lacunae between extra-embryonic mesoderm cells of the fold generate a large cavity closed distally by the amnion, and proximally by the chorion. At LS stage, extra-embryonic mesoderm forms the allantoic bud, precursor to the umbilical cord, in continuity with the primitive streak (*Inman and Downs, 2007*). Extra-embryonic mesoderm cells had striking differences in morphology and migration mode, compared to embryonic mesoderm cells. They were larger and more elongated, displayed fewer filopodia, and almost no large protrusions. They migrated at a similar speed, but in a much more tortuous fashion, resulting in little net displacement.

Direction cues could come from cell-matrix contact, homotypic or heterotypic (with epiblast or visceral endoderm) cell-cell interaction, diffuse gradients of morphogens, and/or mechanical constraints. Transcriptome data were compatible with roles for guidance molecules such as Netrin1 and Eph receptors in directing mesoderm migration. *Epha4* was strongly expressed in the PS and mesoderm, particularly in the embryonic region. In *Xenopus*, interaction of Epha4 in mesoderm and Efnb3

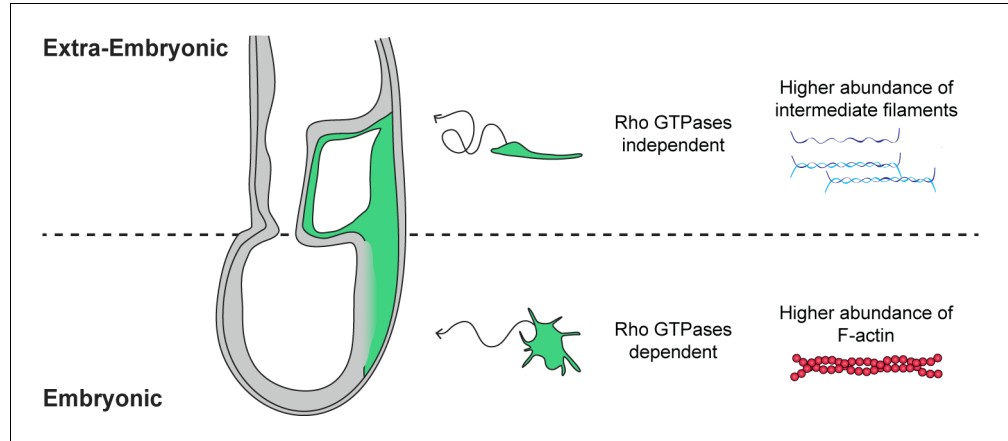

**Figure 7.** Embryonic and extra-embryonic mesoderm cells show distinct shape, trajectory, Rho GTPases dependency, and cytoskeletal composition during gastrulation in the mouse embryo. In the extra-embryonic region (top), mesoderm cells are stretched, with higher Keratin 8 and Vimentin abundance. Their displacement is convoluted and does not depend on Rho GTPases. In the embryonic region (bottom), mesoderm cells are compact with numerous filopodia and have higher F-actin abundance. Cells have straighter trajectories and require Rho GTPases. The embryo scheme represents a sagittal section with anterior to the left. Green and grey label mesoderm and other layers, respectively.
DOI: https://doi.org/10.7554/eLife.42434.044

in ectoderm allows separating germ layers during gastrulation (*Rohani et al., 2014*). *Epha1* and its ligands *Efna1* and *3* had partially overlapping, but essentially reciprocal compartmentalized expression patterns during gastrulation (*Duffy et al., 2006*). In addition, we found abundant *Epha1* expression in somites and presomitic mesoderm at E8.5 (not shown). Interestingly, *Epha1* KO mice present a kinked tail (*Duffy et al., 2008*). The specific and dynamic expression patterns of *Epha4*, *Epha1*, and their respective ligands during gastrulation are compatible with roles in germ layers separation, including nascent mesoderm specification and migration. Identification of those potential guidance cues will help design strategies to better understand how mesoderm subpopulations reach their respective destinations.

Visualization and modification of Rho GTPases activity through FRET sensors and photoactivable variants has shed light on their fundamental role during cell migration in in vivo contexts, such as migration of fish primordial germ cells (*Kardash et al., 2010*) or *Drosophila* border cells (*Wang et al., 2010*). Study of a epiblast-specific mutant showed that Rac1 acts upstream of the WAVE complex to promote branching of actin filaments, lamellipodia formation, and migration of nascent mesoderm (*Migeotte et al., 2011*). Remarkably, extra-embryonic mesoderm cells did not display leading edge protrusions, and *Rac1* and *Rhoa* mesoderm-specific mutants were deficient for embryonic, but not extra-embryonic mesoderm migration. Interestingly, in embryos mutants for *Fgfr1* (*Yamaguchi et al., 1994*) or *Fgf8* (*Sun et al., 1999*), extra-embryonic mesoderm populations are almost normal, while embryonic mesoderm derivatives are severely affected. Measurement of Rho GTPases activity in those mutants would allow exploring the possibility that Rac1 and Rhoa act downstream of the FGF pathway to promote mesoderm migration, as proposed in *Drosophila* (*van Impel et al., 2009*).

In addition, F-actin filaments were more abundant in embryonic, compared to extra-embryonic, mesoderm, reinforcing the hypothesis that they might rely on distinct cytoskeletal rearrangements. Intermediate filaments are major effectors of cell stiffness, cell-matrix and cell-cell adhesion, as well as individual and collective migration (*Pan et al., 2013*). Members of type I and II keratin families form obligate heterodimers, which assemble into filaments (*Loschke et al., 2015*). Type II Keratins 7 and 8, and type I Keratins 18 and 19 are the first to be expressed during embryogenesis. Combined Keratins 8/19 and 18/19 deletions cause lethality at E10 attributed to fragility of giant trophoblast cells (*Hesse et al., 2000*). Deletion of the entire type II Keratins cluster results in growth retardation starting at E8.5 (*Vijayaraj et al., 2009*). Recently, knockdown of Keratin 8 in frog mesendoderm

highlighted a role for intermediate filaments in coordinating collectively migrating cells. Keratin-depleted cells were more contractile, displayed misdirected protrusions and large focal adhesions, and exerted higher traction stress (*Sonavane et al., 2017*). Transcripts for *Keratins 8* and *18*, as well as *Vimentin,* were enriched in extra-embryonic, compared to embryonic mesoderm. While Vimentin was present in all mesoderm, Keratin 8 was only detectable in extra-embryonic mesoderm. An antagonistic relationship between Vimentin intermediate filaments and Rac1-mediated lamellipodia formation has been described (*Helfand et al., 2011*), and a similar opposition may exist between Rac1 activity and keratin intermediate filaments (*Weber et al., 2012*). Extra-embryonic mesoderm cells' elongated morphology, paucity in lamellipodia, and lack of directional migration may thus result from their high content in intermediate filaments, and low Rho GTPase activity (*Figure 7*). The recent development of a K8-YFP reporter mouse strain for intermediate filaments (*Schwarz et al., 2015*), and the availability of reliable Rho GTPases FRET sensors (*Spiering and Hodgson, 2011*), will be instrumental in dissecting their relationship in mesoderm.

The mesoderm germ layer has the particularity to invade both embryonic and extra-embryonic parts of the conceptus, and its migration is important for both fetal morphogenesis and development of extra-embryonic tissues including the placenta. We found that embryonic and extra-embryonic mesoderm populations, both arising by epithelial-mesenchymal transition at the primitive streak, display distinct shape dynamics, migration modes, Rho GTPase dependency, cytoskeletal composition, as well as expression of different sets of guidance, adhesion, and matrix molecules. Landmark experiments in the 1990 s showed that the fate of a mesoderm cell depends on the time and place at which it emerges from the primitive streak. We have unveiled morphological and behavioral specificities of mesoderm populations through whole embryo live imaging, and provided a molecular framework to understand how cells with distinct fates adapt to, and probably modify, their tridimensional environment.

# Materials and methods

**Key resources table**

| Reagent type (species) or resource | Designation | Source or reference | Identifiers | Protocol |
|---|---|---|---|---|
| Genetic reagent (*Mus musculus*) | T-Cre | PMID: 18708576 | RRID:MGI:3811072 | |
| Genetic reagent (*Mus musculus*) | Sox2-Cre | PMID: 12617844; The Jackson Laboratories | TJL: 008454 | |
| Genetic reagent (*Mus musculus*) | mTmG | PMID: 17868096; The Jackson Laboratories | TJL: 007676 | |
| Genetic reagent (*Mus musculus*) | *Rac1* conditional mutant | PMID: 14564011 | RRID:MGI:3579087 | |
| Genetic reagent (*Mus musculus*) | *Rhoa* conditional mutant | PMID: 21209320 | | |
| Genetic reagent (Mus musculus) | LifeAct-mEGFP | PMID: 22658956 | | |
| Antibody | Goat polyclonal anti-Pecam-1 | R and D systems | AF3628 | IF 1/500 |
| Antibody | Rabbit polyclonal anti-Podocalyxin | EMD Millipore | ABD27 | IF 1/200 |
| Antibody | Rat monoclonal anti-Keratin 8 | Developmental Studies Hybridoma Bank | TROMA-I; AB_531826 | IF 1/100 |
| Antibody | Rabbit monoclonal anti-Vimentin | abcam | ab 92547 | IF 1/200 |

*Continued on next page*

*Continued*

| Reagent type (species) or resource | Designation | Source or reference | Identifiers | Protocol |
|---|---|---|---|---|
| Antibody | Donkey polyclonal anti rabbit Alexa Fluor 488 | Life technologies | A21206 | IF 1/500 |
| Antibody | Goat polyclonal anti rabbit Alexa Fluor 647 | Life technologies | A21244 | IF 1/500 |
| Antibody | Chicken polyclonal anti rat Alexa Fluor 647 | Life technologies | A21472 | IF 1/500 |
| Antibody | Donkey polyclonal anti goat Alexa Fluor 647 | Jackson | A21447 | IF 1/500 |
| Other | TRITC-Phalloidin | Invitrogen | A12380 | 1/200 |
| Other | DAPI | Sigma | D9542 | 1/1000 |
| Commercial assay or kit | RNAscope | ACDbio | ACDbio: 322350 | |

## Mouse strains and genotyping

The *T*-Cre line was obtained from Achim Gossler (*Feller et al., 2008*), the *Rac1* line from Victor Tybulewicz (*Walmsley et al., 2003*), the *Rhoa* line from Cord Brakebusch (*Jackson et al., 2011*), the mTmG (*Muzumdar et al., 2007*) and *Sox2*-Cre (*Hayashi et al., 2002*) lines from The Jackson Laboratory, and the conditional LifeAct-GFP line from Laura Machesky (*Schachtner et al., 2012*). Mice were kept on a CD1 background. Mice colonies were maintained in a certified animal facility in accordance with European guidelines. Experiments were approved by the local ethics committee (CEBEA).

Mouse genomic DNA was isolated from ear biopsies following overnight digestion at 55°C with 1.5% Proteinase K (Quiagen) diluted in Lysis reagent (DirectPCR, Viagen), followed by heat inactivation.

## Embryo culture and live imaging

Embryos were dissected in Dulbecco's modified Eagles medium (DMEM) F-12 (Gibco) supplemented with 10% FBS and 1% Penicillin-Streptomycin and L-glutamine and 15 mM HEPES. They were then cultured in 50% DMEM-F12 with L-glutamine without phenol red, 50% rat serum (Janvier), at 37°C and 5% $CO_2$. Embryos were observed in suspension in individual conical wells (Ibidi) to limit drift, under a Zeiss LSM 780 microscope equipped with C Achroplan 32x/0.85 and LD C Apochromat 40x/1.1 objectives. Stacks were acquired every 20 min with 3 µM Z intervals for up to 10 hr. Embryos were cultured for an additional 6 to 12 hr after imaging to check for fitness.

## Antibodies

Antibodies were goat anti-Pecam-1 1:500 (R and D systems); rabbit anti-Podocalyxin 1:200 (EMD Millipore); rat anti-Keratin 8 1:100 (Developmental Studies Hybridoma Bank); rabbit anti-Vimentin 1:200 (abcam). F-actin was visualized using 1.5 U/ml TRITC-Phalloidin (Invitrogen), and nuclei using DAPI (Sigma). Secondary antibodies were anti rabbit Alexa Fluor 488 and 647, anti rat Alexa Fluor 647 (all from Life technologies), and anti goat Alexa Fluor 647 (Jackson).

## Embryo analysis

Whole-mount in situ hybridization was carried out as described in *Eggenschwiler and Anderson (2000)*. For in situ hybridization on sections, embryos were dissected in PBS and fixed for 30 min at 4°C in 4% PFA. They were washed in PBS, embedded directly in OCT (Tissue-Tek), and cryosectioned at 7–10 microns. Slides were re-fixed for 15 min on ice in 4% PFA. RNA probes were obtained from ACDBio, and hybridization was performed using the RNAscope 2.5 HD Reagent Kit-RED

(ACDBio) according to manufacturer's instructions. Slides were counterstained with 50% Gill's Hematoxylin.

For immunofluorescence, embryos were fixed in PBS containing 4% paraformaldehyde (PFA) for 2 hr at 4°C, cryopreserved in 30% sucrose, embedded in OCT and cryosectioned at 7–10 microns. Staining was performed in PBS containing 0.5% Triton X-100% and 1% heat-inactivated horse serum. Sections and whole-mount embryos were imaged on a Zeiss LSM 780 microscope.

## Explant culture and analysis

Primary explant cultures of nascent mesoderm were generated as described in *Burdsal et al. (1993)*. Explants were cultured for 24–48 hr in DMEM F-12 supplemented with 10% FBS and 1% Penicillin-Streptomycin and L-glutamine on fibronectin (Sigma) coated glass bottom microwell 35 mm dishes with 1.5 cover glass (MatTek). They were fixed for 30 min in PBS containing 4% PFA prior to staining. For live imaging, explants were let to adhere for 4–6 hr, and then imaged every 15 min for up to 12 hr.

## Image analysis

Images were processed using Arivis Vision4D v2.12.3 (Arivis, Germany). Embryo contours were segmented manually on each Z-slice and time point, and then registered using the drift correction tool of Arivis Vision4D. Embryo rotation was adjusted manually if necessary. We chose embryos where successful registration could be achieved, so that the embryo's residual slight movements were much smaller than cell displacement. Similarly, we found embryo growth to be negligible compared to cell displacement (data not shown). Cells were then manually segmented on each Z-slice and time point by highlighting cellular membranes using Wacom's Cintiq 13HD.

Net displacement, path length, speed and angle between two cells were based on the centroid coordinates of segmented cells from Arivis, and calculated by a homemade Python script (Python Software Foundation, https://www.python.org). To extract speed behavior, we interpolated the path length curve and derivated it. The path length over time was closely linear, so we extracted the mean of the speed values. Surface, volume, long/short axis ratio of 2D inner ellipse, and straightness were calculated by Arivis. 2D Z projections of late embryos were used to quantify the filopodia length and density. Filopodia size and density were measured on Icy (*de Chaumont et al., 2012*) and analyzed using a homemade Python script.

Videos were generated using the StackReg ImageJ plugin (*Thévenaz et al., 1998*).

All data are presented as Mean ± SEM. Depending on whether data had a Gaussian distribution or not, we used either the Mann-Whitney-Wilcoxon or the *t*-test. A *p* value < 0.05 was considered statistically significant.

## Transcriptome analysis

*T*-Cre; mTmG embryos were collected from different mice, and those at the appropriate stage were pooled. Embryonic and extra-embryonic portions were separated by manually cutting the embryo with finely sharpened forceps. The embryos were digested using 2X Trypsin, and pure GFP + populations were sorted through flow cytometry (FACSARIA III, BD), directly in extraction buffer. RNA was extracted using the PicoPure kit (ThermoFisher Scientific). RNA quality was checked using a Bioanalyzer 2100 (Agilent technologies). Indexed cDNA libraries were obtained using the Ovation Solo RNA-Seq System (NuGen) following manufacturer recommendation. The multiplexed libraries (18 pM) were loaded on flow cells and sequences were produced using a HiSeq PE Cluster Kit v4 and TruSeq SBS Kit v3-HS from a Hiseq 1500 (Illumina). Paired-end reads were mapped against the mouse reference genome (GRCm38.p4/mm10) using STAR software to generate read alignments for each sample. Annotations Mus_musculus.GRCm38.87.gtf were obtained from ftp. Ensembl.org.

For transcript quantification, all the Reference Sequence (RefSeq) transcript annotations were retrieved from the UCSC genome browser database (mm10). Transcripts were quantified using the featureCounts (*Liao et al., 2014*) software tool using the UCSC RefSeq gene annotations (exons only, gene as meta features). Normalized expression levels were estimated using the EdgeR rpm function and converted to log2 FPKM (fragments per kilobase of exon per million mapped reads) after resetting low FPKMs to one to remove background effect. Differential analysis was performed

using the edgeR method (quasi-likelihood tests) (*McCarthy et al., 2012*). The edgeR model was constructed using a double pairwise comparison between embryonic mesoderm versus extra-embryonic mesoderm at two different time points (MS and LS). First, the count data were fitted to a quasi-likelihood negative binomial generalized log-linear model using the R glmQLFit method. To identify differentially expressed genes, null hypothesis EM_E7.0==EEM_E7.0 and EM_E7.25==EEM_E7.25 were tested using the empirical Bayes quasi-likelihood F-tests (glmQLFTest method) applied to the fitted data. The F-test P-values were then corrected for multi-testing using the Benjamini-Hochberg p-value adjustment method. Transcripts with a greater than background level of expression (mean log2 count per million >0), an absolute fold change >2, and a low false discovery rate (FDR <0.05) were considered as differentially expressed.

The sample visualization map was produced by applying the t-Distributed Stochastic Neighbor Embedding (tSNE) dimensionality reduction method (*Van Der Maaten and Hinton, 2008*) to log2 FPKM expression levels (all transcripts). The R tSNE method from 'Rtsne' library was applied without performing the initial PCA reduction and by setting the perplexity parameter to 2. The heatmap was produced using the R heatmap.2 methods using the brewer.pal color palette. GO analysis was performed using the DAVID software (*Huang et al., 2009*).

## Acknowledgments

We thank the animal house, FACS, and light microscopy (LiMiF) core facilities at the ULB (Erasme Campus), and the Brussels Interuniversity Genomics High Throughput core (www.brightcore.be). We thank M Martens and J-M Vanderwinden for confocal imaging help, A Lefort and F Libert for RNA sequencing help, C Brakebusch, A Gossler, V Tybulewicz, and L Machesky for kindly sharing mouse lines, and A Zwijsen, J Bloomekatz, and E Urizar for constructive comments on the manuscript. BS has been sequentially supported by a fellowship from Erasmus Mundus Phoenix and a fellowship of the FRS/FRIA. NM received a fellowship of the FRS/FRIA, as well as support from the 'Fonds David et Alice van Buuren' and the 'Fondation Jaumotte-Demoulin'. WN is supported by WELBIO. IM is a FNRS research associate and an investigator of WELBIO. WELBIO, the FNRS, and the Fondation Erasme supported this work. The authors declare no financial or non-financial competing interests.

## Additional information

### Funding

| Funder | Grant reference number | Author |
|---|---|---|
| Fonds De La Recherche Scientifique - FNRS | PDR T008416 | Isabelle Migeotte |
| Walloon Excellence in Life-sciences and Biotechnology | Welbio CR-2015S-02 | Isabelle Migeotte |
| Fondation Erasme | | Isabelle Migeotte |
| Erasmus Mundus Phoenix | Graduate student fellowship | Bechara Saykali |
| Fonds Alice et David van Buuren | Graduate student fellowship | Navrita Mathiah |
| Fondation Jaumotte-Demoulin | Graduate student fellowship | Navrita Mathiah |
| Fonds De La Recherche Scientifique - FNRS | Graduate student fellowship | Bechara Saykali Navrita Mathiah |

The funders had no role in study design, data collection and interpretation, or the decision to submit the work for publication.

### Author contributions

Bechara Saykali, Conceptualization, Formal analysis, Validation, Investigation, Visualization, Methodology, Writing—review and editing; Navrita Mathiah, Conceptualization, Formal analysis, Investigation, Visualization; Wallis Nahaboo, Conceptualization, Software, Formal analysis, Validation, Investigation, Visualization, Methodology, Writing—review and editing; Marie-Lucie Racu, Latifa

Hammou, Investigation, Visualization; Matthieu Defrance, Formal analysis, Validation, Visualization, Methodology; Isabelle Migeotte, Conceptualization, Resources, Formal analysis, Supervision, Funding acquisition, Validation, Investigation, Visualization, Methodology, Writing—original draft, Project administration, Writing—review and editing

### Author ORCIDs
Bechara Saykali (iD) http://orcid.org/0000-0002-9097-687X
Navrita Mathiah (iD) https://orcid.org/0000-0001-6525-7796
Wallis Nahaboo (iD) https://orcid.org/0000-0001-8283-097X
Matthieu Defrance (iD) https://orcid.org/0000-0002-3090-3142
Isabelle Migeotte (iD) http://orcid.org/0000-0002-8972-8211

### Ethics
Animal experimentation: Animal colonies were maintained in a certified animal facility in accordance with European guidelines. The study protocol was approved by the Commission d'Ethique du Bien-Être Animal (CEBEA) (protocols 394N and 576N)

### Decision letter and Author response
Decision letter https://doi.org/10.7554/eLife.42434.049
Author response https://doi.org/10.7554/eLife.42434.050

## Additional files

### Supplementary files
• Transparent reporting form
DOI: https://doi.org/10.7554/eLife.42434.045

### Data availability
Normalised read counts of the RNASeq data have been deposited in Dryad (doi:10.5061/dryad.8g1nn0j). All other data are included in the manuscript and supporting files. Source Data have been provided for Figures 1, 2, 3, 4 and 6.

The following dataset was generated:

| Author(s) | Year | Dataset title | Dataset URL | Database and Identifier |
|-----------|------|---------------|-------------|--------------------------|
| Saykali B, Mathiah N, Nahaboo W, Racu M, Defrance M, Migeotte I | 2018 | Data from: Distinct mesoderm migration phenotypes in extra-embryonic and embryonic regions of the early mouse embryo | http://dx.doi.org/10.5061/dryad.8g1nn0j | Dryad Digital Repository, 10.5061/dryad.8g1nn0j |

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
