## [Decision Letter]

Thank you for submitting your article "Distinct mesoderm migration phenotypes in extra-embryonic and embryonic regions of the early mouse embryo" for consideration by *eLife*. Your article has been reviewed by three peer reviewers, including Lilianna Solnica-Krezel as the Reviewing Editor and Reviewer #1, and the evaluation has been overseen by Jonathan Cooper as the Senior Editor. The following individual involved in review of your submission has agreed to reveal their identity: Margot Williams (Reviewer #3).

The reviewers have discussed the reviews with one another and the Reviewing Editor has drafted this decision to help you prepare a revised submission.

Using beautiful in vivo imaging, the manuscript describes migratory behaviors of individual internalized embryonic and extraembryonic mesoderm cells in gastrulating mouse embryo, and reveals differences between them. The differences in migration, cell shape, actin organization and protrusive activities between embryonic and extra-embryonic mesoderm (ExM) are correlated with RNA-seq dataset that reveals interesting differences in gene expression.

The authors first provide evidence of reduced speed, displacement, and straightness of ExM migration compared with embryonic mesoderm (EM), and correlate these with quantitative and qualitative differences in cell shape and number and type of cellular protrusions produced. They then profile gene expression in purified ExM and EM populations to identify possible molecular regulators of the distinct cell behaviors they identified. Differentially expressed genes include those with potential roles in cell migration, including adhesion, extracellular matrix, guidance, and cytoskeleton genes. Notably, genes associated with the actin cytoskeleton tend to be reduced in ExM, while those encoding intermediate filaments are increased in ExM.

In light of the observed differences between cytoskeleton and migratory behavior between ExM and EM, the authors evaluate the role of Rho GTPases RhoA and Rac1 in mesoderm migration by conditional loss-of-function in cells traversing the primitive streak. They find that migration of ExM is less affected than EM by loss of both GTPases, demonstrating distinct molecular regulation of these two mesoderm populations. Based on these functional studies, the authors argue that embryonic but not extra-embryonic mesoderm depends on Rho GTPases, indicating reliance on distinct cytoskeletal rearrangements.

It has long been recognized that genetic regulation of ExM migration is distinct from that of embryonic mesoderm, but the cellular and molecular basis of this distinction was largely unknown. This is an interesting, concise and carefully conducted study that sheds light on this issue. The descriptive aspects of the manuscript are novel and should be of interest to *eLife* readership. The RNA-seq data set, in particular for the ExM, provides valuable data for future analyses and intra as well as inter-species comparisons, especially now when aspects of mouse and human gastrulation are studied in synthetic in vitro scenarios. For these reasons, the reviewers feel this work makes important advances in our understanding of gastrulation morphogenesis. However, there are some missed opportunities that could have provided exciting data regarding the relationship between mesodermal cells and surrounding tissues, in particular, the visceral endoderm. There are additional issues (listed below) that the authors should address before this report is published.

Essential revisions:

1) It has been shown that the proximal third of the visceral endoderm cells overlying the epiblast in the posterior region remain relatively coherent forming a crescent that tapers anteriorly (Kwon et al., 2008). This region has been proposed by Kwon and co-workers to be a signaling center. This area of visceral endoderm appears to overlay the area of spread of embryonic mesodermal cells in the proximal posterior side of the embryo. Is there a relationship between the movements of posterior mesoderm and posterior visceral endoderm? This issue should be at least discussed by the authors, if addressing it experimentally would extend the revision beyond 2 months.

2) Kwon et al. (2008) also argued that distal visceral endoderm cells intercalate with definitive endoderm cells. Was this phenomenon evident in the authors' experiments?

3) The laboratory of Karen Downs has argued that posterior visceral endoderm cells located in the extra-embryonic region exfoliate and contribute to extra-embryonic mesoderm derivatives like the allantois (Rodriguez and Downs, 2017). Was this phenomenon observed in the present studies? Are GTPases not required for extraembryonic mesoderm migration, or simply are there different genetic redundancies between the two cell types? There should be at least some discussion regarding upstream regulators of Rho/Rac1 in embryonic mesoderm.

4) Several figures contain graphs depicting the number of filopodia per cell. Is this filopodia per unit time? Or at a single time point? It just seems unrealistic that a mesoderm cell would produce only 7 filopodia over the course of several hours of imaging. This could be the result of the imaging interval, though. Cell protrusions are extremely dynamic, and one stack every 20 minutes likely does not provide sufficient temporal resolution required for this type of quantification. Could the authors please clarify how such measurements were made?

5) In gene expression profiling experiments, EM and ExM samples are compared to "non-mesoderm controls" – what types of tissues are these? And why do they cluster with embryonic mesoderm but not the ExM?

6) When describing the results of these profiling experiments, the authors discuss the enrichment of certain GO terms and gene clusters among differentially expressed genes, but it is not clear whether genes associated with these terms/clusters exhibit increased or decreased expression in ExM vs. EM. This is even the case in Figure 4—figure supplement 1A, where the p value of each enriched term is presented, but no information is given about which is increased or decreased in ExM or EM.

7) A major conclusion of this report is that "extra-embryonic mesoderm movement was… GTPase dependent". However, no quantification is provided in support of this critical finding. Although ExM cells can be seen in the extra-embryonic region of RhoA and Rac1; T-cre conditional KO embryos, it was not determined whether the number of these cells is comparable to control embryos, and whether they exhibit similar dynamics during migration. The authors have demonstrated their ability to quantify size and number of protrusions, migration speed, persistence etc. of migrating mesoderm cells. Cell trajectories and migration speed of ExM cells should be analyzed for Rac1 and RhoA mutant ExM cells and compared to wild type, as done in Table 1.

8) The T(s)-Cre transgenic line was used to drive cre expression in the primitive streak. In this line, cre expression is driven by the Brachyury (T) promoter. T, however, is also expressed in extra-embryonic ectoderm (Rivera-Perez and Magnuson, 2005; Perea-Gomez et al., 2004). I assume that in T(s)-cre embryos, cre is not expressed in the extra-embryonic ectoderm. If this is the case, the authors need to stress that the transgene is only expressed in the primitive streak. Also, please stress that it is a random insertion transgenic line not a knock-in line into the T locus that inactivates T.

9) "Posterior views (Figure 1A) showed proximal to distal primitive streak extension and rounding of cells exiting the streak". Epiblast cells are elongated in apical-basal direction. In this image, indeed shapes of the cells in the midline appear round. It is not clear however, whether this represents a cross-section of elongated cells or a true round shape in 3D. Additional images or information should be provided to support this conclusion. In addition, is this confirming earlier observations or a novel observations of the authors?

10) "Extra-embryonic cells were stretched, sometimes spanning the entire widths of the embryo, and twice larger". Could the authors comment on their third dimension? Were they flatter compared to embryonic cells?

It is also striking that these extraembryonic cells have twice the volume of embryonic mesodermal cells (Table 2). What is the reason for this difference? Are the precursors in the primitive streak different? Or fewer cell divisions in the extra-embryonic population?

11) In general, the manuscript would be improved by clearly stating the questions motivating individual lines of experiments and providing conclusions. For example, the title of this section is "Cell-cell communication within the mesodermal layer", yet no question is posed. The results describe relative migratory behaviors of daughter cells in mesodermal cell layer. In the second part of this section the behavior of mesodermal cells upon contact that usually not result in contact inhibition is contrasted to that previously reported for neural crest cells. One concern about this section is why it is titled "cell-cell communication" as there is no direct evidence for communication between cells apart from cells migrating "in the same direction, remaining close to one another, and staying linked by thin projections." Second, the behavior of daughter cells and cells close to one another at the beginning of a time-lapse should be compared. Is the described behavior of two daughter cells unusual or is it typical for two cells that happen to be close to one another? Finally, there is no conclusion at the end of the section.

12) The findings on the migratory cell behaviors of mesodermal cells should be also better presented in the context of gastrulation studies in other systems and the work in the mouse. Whereas the authors emphasize collective cell behaviors in *Drosophila* and other vertebrates, meandering migration of mesodermal cells was documented in the *Fundulus* fish by pioneering studies of John Trinkaus and in zebrafish by studies from Solnica-Krezel laboratory, as well as in the chick embryo by studies from Weijer laboratory. In particular the meandering directed migration of individual mesodermal cells with protrusive activity is highly similar to that described by Jessen et al., 2002 in the zebrafish gastrula. Therefore, Migeotte and colleagues demonstrate that in mouse embryo, mesoderm gastrulation movements entail directed migration on embryonic mesodermal cells in addition to the well described intercalation movements of axial mesoderm.

Furthermore, while reflecting upon differences between regulation of embryonic and extraembryonic mesoderm migration, it may be helpful to note results from previous studies that also observed critical differences in these two populations:

A) Embryonic mesoderm fails to migrate away from the PS in FGF8 and FGFR1 mutant mouse embryos, but extraembryonic mesoderm can migrate away (Yamaguchi et al., 1994; Sun et al., 1999).

B) Mutations affecting BMP signaling, including BMP4, Pace/Furin, and BMP receptors prevent formation of an embryonic primitive streak, but allow for production of some extraembryonic mesoderm (Gu et al., 1999; Song et al., 1999; Beck et al., 2002; Winnier et al., 1995).

Also, the authors note on the presence of tumbling behavior alternating with straighter displacement, which is strikingly similar to "tumble and run" movements described in zebrafish embryos (Diz-Munoz et al., 2016). This similarity is worth noting and citing.

References:

Perea-Gomez A, Camus A, Moreau A, Grieve K, Moneron G, Dubois A, Cibert C, Collignon J. Initiation of gastrulation in the mouse embryo is preceded by an apparent shift in the orientation of the anterior-posterior axis. Curr Biol. 2004 Feb 3;14(3):197-207.

Rivera-Pérez JA, Magnuson T. Primitive streak formation in mice is preceded by localized activation of Brachyury and Wnt3. Dev Biol. 2005 Dec 15;288(2):363-71.

Rodriguez AM, Downs KM. Visceral endoderm and the primitive streak interact to build the fetal-placental interface of the mouse gastrula. Dev Biol. 2017 Dec 1;432(1):98-124. doi: 10.1016/j.ydbio.2017.08.026.

[Editors' note: further revisions were requested prior to acceptance, as described below.]

Thank you for resubmitting your work entitled "Distinct mesoderm migration phenotypes in extra-embryonic and embryonic regions of the early mouse embryo" for further consideration at *eLife*. Your revised article has been favorably evaluated by Jonathan Cooper as the Senior Editor and a Reviewing Editor.

In this revised manuscript the authors largely addressed the reviewers' questions and concerns. The manuscript significantly advances our understanding of cellular and molecular mechanisms of mesoderm migration during mouse gastrulation, and reveals interesting cellular and molecular mechanisms between embryonic and extraembryonic mesoderm migration. Whereas the manuscript has been improved, a few remaining issues need to be addressed before acceptance, as outlined below:

As suggested by the reviewers, the authors analyzed movements of pairs of neighboring cells (subsection “Impact of cell-cell contact on trajectory within the mesoderm layer”, third paragraph). To make clear the difference between this scenario and the scenario of two daughter cells that migrated in concert, would it be correct to state "Pairs of unrelated cells that were in immediate proximity"? Also have the authors observed any projections linking such cells as they observed for daughter pairs (see the second paragraph of the aforementioned subsection)? This is stated later in the concluding sentence, but should be stated here.

Moreover, the observation that unrelated cells can also follow parallel paths should be included in the third paragraph of the Discussion.

Please indicate developmental stages on the figure panels.

Subsection “Extra-embryonic mesoderm migration is Rho GTPases independent”, third paragraph: "…live imaging of Rhoa deleted explants" rather "live imaging of explants from Rhoa mutant gastrulae or embryos".

---

## [Author Response]

Essential revisions:1) It has been shown that the proximal third of the visceral endoderm cells overlying the epiblast in the posterior region remain relatively coherent forming a crescent that tapers anteriorly (Kwon et al., 2008). This region has been proposed by Kwon and co-workers to be a signaling center. This area of visceral endoderm appears to overlay the area of spread of embryonic mesodermal cells in the proximal posterior side of the embryo. Is there a relationship between the movements of posterior mesoderm and posterior visceral endoderm? This issue should be at least discussed by the authors, if addressing it experimentally would extend the revision beyond 2 months.

Visceral endoderm cells are marked with membrane Tomato. We designed the condition of two-photon live imaging in order to get an optimal GFP signal, and just enough Tomato to get the anatomy of the embryo, to reduce phototoxicity. Due to the relatively weak Tomato signal and the morphology of the posterior visceral endoderm cells, which tend to be rather flat, we were not able to reliably track visceral endoderm cells movements.

We added a reference to the potential signaling function of the posterior visceral endoderm proposed by Kwon et al. (subsection “Mesoderm cells have distinct morphology depending on their interaction with different germ layers”).

2) Kwon et al. (2008) also argued that distal visceral endoderm cells intercalate with definitive endoderm cells. Was this phenomenon evident in the authors' experiments?

Cells crossing the streak to become definitive endoderm do not appear to express Cre, as we very rarely observed GFP positive cells in the endoderm layer, even when looking at later time points (E7.75).

The transgenic T(s)::Cre line was engineered using a construct composed of regulatory elements of the Brachyury gene directing gene expression in the primitive streak fused to a Cre cDNA. It does not recapitulate all aspects of endogenous T expression. This observation fits with the characterization of the line published in Feller et al., 2008: in Figure 1—figure supplement 1A, a T(s)::Cre; Rosa26-LacZ embryo at head fold stage has no visible blue cells in the endoderm.

Alternatively, Burtscher and Lickert (2009) described anti-parallel gradients of FoxA2 and T along the streak, so it is possible that T might not label endoderm progenitors, just midline and mesoderm. We have not done the experiment to distinguish those two possibilities.

We clarified the design of the T-Cre transgenic line (subsection “Mesoderm migration mode and cell shape differ in embryonic versus extra-embryonic regions”, first paragraph), stated that we did not visualize definitive endoderm, and proposed explanations (subsection “Mesoderm cells have distinct morphology depending on their interaction with different germ layers”).

3) The laboratory of Karen Downs has argued that posterior visceral endoderm cells located in the extra-embryonic region exfoliate and contribute to extra-embryonic mesoderm derivatives like the allantois (Rodriguez and Downs, 2017). Was this phenomenon observed in the present studies? Are GTPases not required for extraembryonic mesoderm migration, or simply are there different genetic redundancies between the two cell types? There should be at least some discussion regarding upstream regulators of Rho/Rac1 in embryonic mesoderm.

As mentioned above, Tomato expressing cells could not be visualized with as much clarity, which rendered their tracking unreliable. In addition, experiments suggesting delamination of posterior visceral endoderm cells to become extra-embryonic mesoderm were mostly conducted from the Zero Bud stage onwards, while most of our recordings were performed between Early Streak and Zero Bud stages, so we would not have been in the best position to observe this phenomenon.

Indeed, our data cannot differentiate between absence of dependency on RhoGTPases, and different redundancy. We have corrected this in the manuscript (subsection “Extra-embryonic mesoderm migration is Rho GTPases independent”, last paragraph). We have also proposed that RhoGTPases might be downstream of Fgf signaling (Discussion, sixth paragraph).

4) Several figures contain graphs depicting the number of filopodia per cell. Is this filopodia per unit time? Or at a single time point? It just seems unrealistic that a mesoderm cell would produce only 7 filopodia over the course of several hours of imaging. This could be the result of the imaging interval, though. Cell protrusions are extremely dynamic, and one stack every 20 minutes likely does not provide sufficient temporal resolution required for this type of quantification. Could the authors please clarify how such measurements were made?

This was indeed confusing. It is "filopodia per cell per time point". It has been clarified in the legends.

5) In gene expression profiling experiments, EM and ExM samples are compared to "non-mesoderm controls" – what types of tissues are these? And why do they cluster with embryonic mesoderm but not the ExM?

The non-mesoderm controls contain a mixture of all non-GFP cells, which include extra-embryonic ectoderm, epiblast, visceral endoderm, and probably some non-labeled mesoderm cells. We did not analyze those data in detail, as results were quite heterogeneous between samples, likely due to unequal proportions of cells from each population in different samples, as we did not collect all non-GFP cells from each pool of embryos. Embryonic controls tend to be closer to embryonic mesoderm indeed, probably because the majority of the cells are likely to be of epiblast origin. Since data from those samples were not discussed, we eliminated them from the tSNE plot and Table 3.

6) When describing the results of these profiling experiments, the authors discuss the enrichment of certain GO terms and gene clusters among differentially expressed genes, but it is not clear whether genes associated with these terms/clusters exhibit increased or decreased expression in ExM vs. EM. This is even the case in Figure 4—figure supplement 1A, where the p value of each enriched term is presented, but no information is given about which is increased or decreased in ExM or EM.

GO had been performed based on the list of differentially regulated genes, whichever the direction. The QLF list has now been divided in two according to the region of higher expression, and GO has been performed based on those two lists. This analysis is displayed in Figure 4—figure supplement 1A, and describe in the text (subsection “Embryonic and extra-embryonic mesoderm molecular signatures”, first paragraph).

7) A major conclusion of this report is that "extra-embryonic mesoderm movement was… GTPase dependent". However, no quantification is provided in support of this critical finding. Although ExM cells can be seen in the extra-embryonic region of RhoA and Rac1; T-cre conditional KO embryos, it was not determined whether the number of these cells is comparable to control embryos, and whether they exhibit similar dynamics during migration. The authors have demonstrated their ability to quantify size and number of protrusions, migration speed, persistence etc. of migrating mesoderm cells. Cell trajectories and migration speed of ExM cells should be analyzed for Rac1 and RhoA mutant ExM cells and compared to wild type, as done in Table 1.

In order to consolidate the data pointing to the difference of dependency on GTPases, we have added quantitative analysis of the live imaging data.

The number of GFP positive cells at the early stages of gastrulation, when tracking of cells shape and speed could be done with precision, was quite variable among wild-type embryos, which we attribute to the mosaicism of T-Cre expression. Comparing cell numbers between wild-type and mutant embryos was thus not practical.

We could however segment cells in mutant embryos, and record quantitative data on cell shape and movements. We segmented mesoderm cells in the embryonic and extra-embryonic regions of mutants. It was difficult to perfectly register mutant embryos due to their irregular shape. Because of the relatively low number of cells available, we tracked them for as long as possible, which resulted in different times of observation. Therefore, we could not directly compare net and travel displacements. We could nonetheless observe a significant decrease in embryonic, but not extra-embryonic mesoderm cells speed for both Rho and Rac mutants, compared to wild-type embryos. The data has been added to Figure 6—figure supplement 1 (Figure 6—figure supplement 1—source data 1 and 2), and is described in the manuscript (subsection “Extra-embryonic mesoderm migration is Rho GTPases independent”, last paragraph).

In addition, we performed live imaging of a Rac1fl/ko; mTmG; Sox2-Cre embryo, and confirmed the presence of GFP positive extra-embryonic mesoderm cells dispersed in the extra-embryonic region, while GFP positive mesoderm cells were confined in a bulge adjacent to the primitive streak in the embryonic region (subsection “Extra-embryonic mesoderm migration is Rho GTPases independent”, last paragraph). Those images have been added to Figure 6—figure supplement 2.

8) The T(s)-Cre transgenic line was used to drive cre expression in the primitive streak. In this line, cre expression is driven by the Brachyury (T) promoter. T, however, is also expressed in extra-embryonic ectoderm (Rivera-Perez and Magnuson, 2005; Perea-Gomez et al., 2004). I assume that in T(s)-cre embryos, cre is not expressed in the extra-embryonic ectoderm. If this is the case, the authors need to stress that the transgene is only expressed in the primitive streak. Also, please stress that it is a random insertion transgenic line not a knock-in line into the T locus that inactivates T.

Please refer to response to point 2. We have examined mTmG; T-cre embryos between E6.25 and E6.5, and have not identified any GFP positive cells in the extra-embryonic ectoderm.

9) "Posterior views (Figure 1A) showed proximal to distal primitive streak extension and rounding of cells exiting the streak". Epiblast cells are elongated in apical-basal direction. In this image, indeed shapes of the cells in the midline appear round. It is not clear however, whether this represents a cross-section of elongated cells or a true round shape in 3D. Additional images or information should be provided to support this conclusion. In addition, is this confirming earlier observations or a novel observations of the authors?

It represents a cross-section of bottle shape cells in their round portion. This confirms earlier observation, in particular those of Williams et al. It has been made clear in the text (subsection “Mesoderm migration mode and cell shape differ in embryonic versus extra-embryonic regions”, second paragraph).

10) "Extra-embryonic cells were stretched, sometimes spanning the entire widths of the embryo, and twice larger". Could the authors comment on their third dimension? Were they flatter compared to embryonic cells? It is also striking that these extraembryonic cells have twice the volume of embryonic mesodermal cells (Table 2). What is the reason for this difference? Are the precursors in the primitive streak different? Or fewer cell divisions in the extra-embryonic population?

Extra-embryonic cells were larger and more elongated. The increase in volume was larger than the increase in surface, which indicates they were not flatter.

We quantified cell divisions in time-lapse recordings. Embryonic and extra-embryonic mesoderm cells were observed over an average of 360 min from 4 different embryos at Mid Streak stage and division events were recorded. Cells migrating close to embryonic/extra-embryonic interface were not taken into consideration to avoid misallocation. Division events counts were normalized over total number of GFP positive cells per region. We found indeed a higher frequency of division in embryonic, compared to extra-embryonic mesoderm. This data has been added to the text (subsection “Mesoderm migration mode and cell shape differ in embryonic versus extra-embryonic regions”, fourth paragraph) and is shown in Figure 1—figure supplement 1 (Figure 1—figure supplement 1—source data 1).

11) In general, the manuscript would be improved by clearly stating the questions motivating individual lines of experiments and providing conclusions. For example, the title of this section is "Cell-cell communication within the mesodermal layer", yet no question is posed. The results describe relative migratory behaviors of daughter cells in mesodermal cell layer. In the second part of this section the behavior of mesodermal cells upon contact that usually not result in contact inhibition is contrasted to that previously reported for neural crest cells. One concern about this section is why it is titled "cell-cell communication" as there is no direct evidence for communication between cells apart from cells migrating "in the same direction, remaining close to one another, and staying linked by thin projections." Second, the behavior of daughter cells and cells close to one another at the beginning of a time-lapse should be compared. Is the described behavior of two daughter cells unusual or is it typical for two cells that happen to be close to one another? Finally, there is no conclusion at the end of the section.

We have modified the subsection title (“Impact of cell-cell contact on trajectory within the mesoderm layer”), and added an introduction and conclusion to the paragraph.

Pairs of cells in immediate close proximity were segmented and tracked. In the embryonic region, paired mesoderm cells traveled a similar distance over 152 min (net displacement ratio: 0.88 ± 0.03, n=13 pairs from 4 embryos), in the same direction (angle: 9.9° ± 2.77). Travel displacement ratio and final distance between paired cells were 0.75 ± 0.04 and 15.2 µm ± 3.84 µm, respectively (Figure 3—source data 3). All values were similar to what was observed for daughter cells displacements, so indeed the behavior of daughter cells resembles that of two cells that happen to be close to one another. This data is included in the third paragraph of the subsection “Impact of cell-cell contact on trajectory within the mesoderm layer”.

12) The findings on the migratory cell behaviors of mesodermal cells should be also better presented in the context of gastrulation studies in other systems and the work in the mouse. Whereas the authors emphasize collective cell behaviors in *Drosophila* and other vertebrates, meandering migration of mesodermal cells was documented in the Fundulus fish by pioneering studies of John Trinkaus and in zebrafish by studies from Solnica-Krezel laboratory, as well as in the chick embryo by studies from Weijer laboratory. In particular the meandering directed migration of individual mesodermal cells with protrusive activity is highly similar to that described by Jessen et al., 2002 in the zebrafish gastrula. Therefore, Migeotte and colleagues demonstrate that in mouse embryo, mesoderm gastrulation movements entail directed migration on embryonic mesodermal cells in addition to the well described intercalation movements of axial mesoderm.

References to the work of John Trinkaus as well as Jessen et al. have been added in the Introduction, and more details were given regarding the work of the Weijer lab (Introduction, third paragraph).

Furthermore, while reflecting upon differences between regulation of embryonic and extraembryonic mesoderm migration, it may be helpful to note results from previous studies that also observed critical differences in these two populations:A. Embryonic mesoderm fails to migrate away from the PS in FGF8 and FGFR1 mutant mouse embryos, but extraembryonic mesoderm can migrate away (Yamaguchi et al., 1994; Sun et al., 1999).

The phenotypes are indeed strikingly similar and should have been referred to. We corrected this in the Discussion (Discussion, sixth paragraph).

B. Mutations affecting BMP signaling, including BMP4, Pace/Furin, and BMP receptors prevent formation of an embryonic primitive streak, but allow for production of some extraembryonic mesoderm (Gu et al., 1999; Song et al., 1999; Beck et al., 2002; Winnier et al. 1995).

In the ActRIA mutant (Gu et al., 1999), there is no primitive streak, and some mesenchymal cells in the extra-embryonic region (identified by morphology, not markers). In the double ActRIIA/IIB mutant there is no streak and no mesoderm, and in the IIA^-/-^IIb^+/-^ there is some mesoderm in both region (Song et al., 1999). In the BMP4 KO (Winnier et al., 1995), there is also no streak, and some small amount of extra-embryonic mesoderm (also identified by morphology). We are afraid the comparison might be confusing here, as due to absence of the primitive streak the phenomenon is likely to differ. The extra-embryonic mesoderm cells observed could arise from another layer (cf. Rodriguez and Downs, 2017). Furin/Pace 4 double mutants are reduced in size at E7.5, there is no amnion, allantois or chorion, and some mesoderm expressing FgF8 and Lhx1 bulges into the cavity, but markers of extra-embryonic mesoderm are not shown (Beck et al., 2002).

Therefore we would rather not include those references.

Also, the authors note on the presence of tumbling behavior alternating with straighter displacement, which is strikingly similar to "tumble and run" movements described in zebrafish embryos (Diz-Munoz et al., 2016). This similarity is worth noting and citing.

Reference to the work of Diz-Munoz et al. has been added (subsection “Mesoderm migration mode and cell shape differ in embryonic versus extra-embryonic regions”, third paragraph).

[Editors' note: further revisions were requested prior to acceptance, as described below.]

In this revised manuscript the authors largely addressed the reviewers' questions and concerns. The manuscript significantly advances our understanding of cellular and molecular mechanisms of mesoderm migration during mouse gastrulation, and reveals interesting cellular and molecular mechanisms between embryonic and extraembryonic mesoderm migration. Whereas the manuscript has been improved, a few remaining issues need to be addressed before acceptance, as outlined below:As suggested by the reviewers, the authors analyzed movements of pairs of neighboring cells (subsection “Impact of cell-cell contact on trajectory within the mesoderm layer”, third paragraph). To make clear the difference between this scenario and the scenario of two daughter cells that migrated in concert, would it be correct to state "Pairs of unrelated cells that were in immediate proximity"? Also have the authors observed any projections linking such cells as they observed for daughter pairs (see the second paragraph of the aforementioned subsection)? This is stated later in the concluding sentence, but should be stated here.

The proposed change in terminology for cells in proximity has been made (subsection “Impact of cell-cell contact on trajectory within the mesoderm layer”, first paragraph). In two-photon videos, we could indeed observe contact between protrusions from cells close to one another. Some of the analyzed pairs are from confocal videos with lesser resolution; for those we cannot say for sure. Nonetheless, we could observe points of contact for 10/12 pairs and have thus included a sentence in that direction (subsection “Impact of cell-cell contact on trajectory within the mesoderm layer”, third paragraph).

Moreover, the observation that unrelated cells can also follow parallel paths should be included in the third paragraph of the Discussion.

A sentence has been included in the Discussion (third paragraph).

Please indicate developmental stages on the figure panels.

This has been done in all figures. In view of their abnormal development, no stage was indicated for mutant embryos; however, the stage of the wild-type littermate displayed in each panel was noted.

Subsection “Extra-embryonic mesoderm migration is Rho GTPases independent”, third paragraph: "…live imaging of Rhoa deleted explants" rather "live imaging of explants from Rhoa mutant gastrulae or embryos".

This has been fixed.